# *C. elegans* enteric motor neurons fire synchronized action potentials underlying the defecation motor program

Jingyuan Jiang [1,7], Yifan Su[1,7], Ruilin Zhang [1,2], Haiwen Li[3], Louis Tao [1,4,8] & Qiang Liu [5,6,8 ✉]

*C. elegans* neurons were thought to be non-spiking until our recent discovery of action potentials in the sensory neuron AWA; however, the extent to which the *C. elegans* nervous system relies on analog or digital coding is unclear. Here we show that the enteric motor neurons AVL and DVB fire synchronous all-or-none calcium-mediated action potentials following the intestinal pacemaker during the rhythmic *C. elegans* defecation behavior. AVL fires unusual compound action potentials with each depolarizing calcium spike mediated by UNC-2 followed by a hyperpolarizing potassium spike mediated by a repolarization-activated potassium channel EXP-2. Simultaneous behavior tracking and imaging in free-moving animals suggest that action potentials initiated in AVL propagate along its axon to activate precisely timed DVB action potentials through the INX-1 gap junction. This work identifies a novel circuit of spiking neurons in *C. elegans* that uses digital coding for long-distance communication and temporal synchronization underlying reliable behavioral rhythm.

[1] Center for Bioinformatics, National Laboratory of Protein Engineering and Plant Genetic Engineering, School of Life Sciences, Peking University, Beijing 100871, China. [2] Yuanpei College, Peking University, Beijing 100871, China. [3] LMAM, School of Mathematical Sciences, Peking University, Beijing 100871, China. [4] Center for Quantitative Biology, Peking University, Beijing 100871, China. [5] Lulu and Anthony Wang Laboratory of Neural Circuits and Behavior, The Rockefeller University, New York, NY 10065, USA. [6] Present address: Department of Neuroscience, City University of Hong Kong, Kowloon, Hong Kong SAR. [7] These authors contributed equally: Jingyuan Jiang, Yifan Su. [8] These authors jointly supervised this work: Louis Tao, Qiang Liu. ✉email: qiangliu@cityu.edu.hk

The ancient enteric nervous system plays a role not only in digestion, but in the broader regulation of physiology and behavior across the animal kingdom. For example, studies of the crustacean stomatogastric nervous system have revealed numerous insights into the cellular and circuit mechanisms that generate rhythmic motor patterns[1]. The intestine and a simple enteric nervous system underlying the *C. elegans* defecation have been a powerful model system to study gut-brain communication and ultradian rhythms. This rhythmic defecation behavior, called the defecation motor program (DMP), consists of a series of stereotyped motor sequences activated once every ~45 s[2]. At the behavioral level, the three distinct, sequential motor steps are referred to as pBoc (posterior Body muscle contraction), aBoc (anterior Body muscle contraction), and Exp (Expulsion, or enteric muscle contraction)[3]. At the anatomical level, these motor steps are driven by a simple gut-brain circuit consisting of the intestine, two enteric motor neurons called AVL and DVB[4], and downstream contractile apparatus including body-wall muscles and enteric muscles specialized in expulsion[3]. Extensive studies using genetic screening, calcium imaging, and behavioral analysis have collectively brought forth the following model: propagating calcium waves in the intestine, mediated by inositol triphosphate (IP3) receptors, generate intrinsic rhythmicity as the pacemaker of the system[5–7]; periodic calcium-mediated release of protons from the posterior intestine directly activates the contraction of posterior body-wall muscles[8]; the intestinal rhythmicity is relayed to the enteric neurons via calcium-mediated release of a neuropeptide-like protein NLP-40 from the intestine, activating the AEX-2/G protein-coupled receptor (GPCR) on AVL and DVB[9,10]; activated AVL and DVB, synchronized by electrical synapses INX-1[11], release the neurotransmitter GABA at the neuromuscular junctions (NMJs) to activate downstream enteric muscles through excitatory GABA receptors EXP-1[12]; coordinated contraction and relaxation of enteric muscle groups finally lead to discrete expulsion events, expelling the gut contents[3]. Despite much progress in the understanding of calcium oscillations, rhythm regulation, and peptidergic signaling made over a span of 30 years by studying the *C. elegans* defecation cycle[2], the biophysics and activity dynamics of enteric neurons and their contribution to the time-keeping mechanism of the behavioral rhythm is not well understood.

The rhythmicity, synchronization, and long-distance communication featured in the *C. elegans* defecation behavior suggest digital signaling that is best served by spiking neurons. Indeed, at least three lines of evidence from previous studies indicate that AVL and DVB may possess all-or-none properties. First, calcium transients observed in AVL and DVB preceding expulsion are discrete with uniform amplitude and kinetics[9,11,13]. Second, genetic manipulations of the defecation circuit almost always affect AVL and DVB activities in an all-or-none fashion, i.e., they may eliminate or restore neuronal activity or in some cases affect the timing, but not the magnitude, of the calcium transients[9,11,13]. Third, injection of neuropeptide NLP-40 into the tail region evoked DVB calcium transients that are indistinguishable from spontaneously occurring ones[9]. Because it is unlikely that exogenous injection would deliver the precise amount of neuropeptides in the correct time scale as the endogenously released counterpart, these results strongly suggest that DVB is gated by an all-or-none thresholding mechanism.

Action potentials digitalize continuous inputs into precisely timed spikes and are crucial for neural coding and signal propagation in most known nervous systems. The *C. elegans* nervous system was thought to be an exception with only graded signaling until the recent discovery of the first neuronal action potentials in an olfactory neuron, AWA[14]. Thus, the *C. elegans* nervous system employs a hybrid system for information processing[15–25], but it is not clear to what extent the digital and analog coding schemes are utilized across different neural cell types or neural functions. The all-or-none property and temporal precision inherently associated with the action potential excel at synchronizing spatially distributed activity, relaying information across distances without decay, and setting time windows for neural oscillation and plasticity. Here we directly examine the intrinsic electrophysiological properties of AVL and DVB and identify both neurons as spiking motor neurons that fire all-or-none action potentials in *C. elegans*. Through recordings from animals with loss- or gain-of-function mutations in various ion channels, we define the ionic basis and the molecular mechanisms underlying these action potentials. Simultaneous imaging of both neurons and the intestine in free-moving animals suggest that the action potential in AVL not only functions in long-distance signal propagation but its specialized features also set the timing for neural synchronization that is important for expulsion behavior. These newly discovered spiking neurons reveal unanticipated heterogeneity in the *C. elegans* nervous system and call for future in-depth investigations of the significance of its biophysical diversity in neural computation and animal behavior.

## Results

**Enteric neurons AVL and DVB fire action potentials with distinct kinetics**. The *C. elegans* enteric nervous system consists of two GABAergic motor neurons, AVL in the lateral ganglion and DVB in the dorso-rectal ganglion. Cell-killing experiments indicate that they drive the expulsion step of the defecation cycle[4,11]. Since expulsion mediated by synchronized action of multiple enteric muscles is an episodic, all-or-none behavior, we suspected that AVL and DVB may possess specialized electrophysiological properties best suited for this task. Thus, we recorded from cell bodies of AVL and DVB in dissected worms with GFP expressed in all GABA neurons using a whole-cell patch-clamp to determine their intrinsic electrophysiological properties and underlying ion channel kinetics. Under the current-clamp recording configuration, all-or-none action potentials were evoked in both AVL and DVB by applying a series of current-injection stimuli (Fig. 1a, b). Several motor neurons have been electrophysiologically characterized including VA and VB[19,20], VD, and DD (unpublished data) neurons but none were found to fire action potentials. Thus, AVL and DVB are the first two spiking motor neurons identified in *C. elegans* to date. In comparison, neither RIS, another GABAergic neuron located in close proximity to AVL in the head region, nor DVC, another neuron located right next to DVB in the tail region, fired action potentials under any current-injection stimuli we tested (Fig. S1a, b). These results suggest that action potential firing is not a general property for GABAergic neurons, nor dictated by the neuron's physical location in the body, but most likely a specialized feature for the two enteric neurons with specific functions in the defecation behavior.

The kinetics of action potentials in AVL and DVB were distinct from each other, as well as from those in AWA[14]. In particular, under suprathreshold stimuli, DVB membrane potential depolarized gradually at first, then evolved into an abrupt upstroke overshooting 20 mV after reaching a threshold of about −20 mV (Fig. 1a). The peak of the upstroke was followed by a fast downstroke and a modest afterhyperpolarization (AHP), forming a broad spike of nearly 400 ms in half-maximum width (Fig. 1a, c). DVB usually fired single or double spikes with similar threshold, amplitude, spike width, and overall shape, the defining features of all-or-none action potentials. Similarly, AVL also fired broad action potential spikes but with smaller peak amplitude (Fig. 1a, c). More strikingly, however, the AHP following each

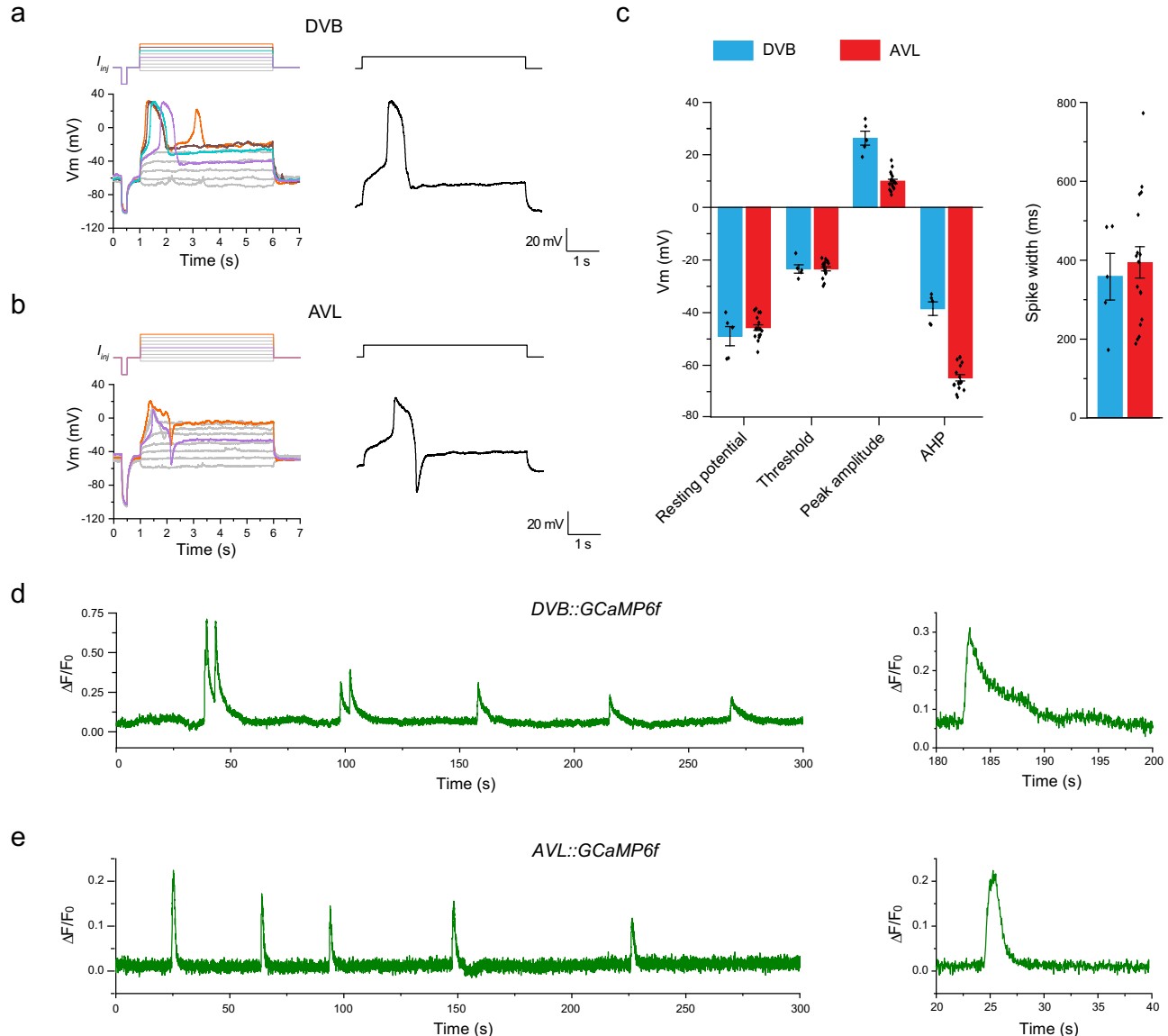

**Fig. 1 Enteric neurons AVL and DVB fire action potentials with distinct kinetics. a, b** Representative current-clamp recording traces from DVB and AVL. Top: current-injection stimulation steps (5 s long at 1 pA increment). Left: membrane potential dynamics under color-matched stimulation steps. Right: representative single recording traces with action potentials fired. **c** Statistical measurements of resting membrane potential (DVB: −49.0 ± 3.6 mV; AVL: −45.7 ± 1.1 mV), threshold (DVB: −23.4 ± 1.6 mV; AVL: −23.3 ± 0.8 mV), peak amplitude (DVB: 26.3 ± 2.6 mV; AVL: 9.8 ± 0.8 mV), afterhyperpolarization (AHP) (DVB: −38.5 ± 2.5 mV; AVL: −64.8.3 ± 1.2 mV), and half-maximum spike width (DVB: 358.8 ± 59.5 ms; AVL: 394.2 ± 40.0 ms) for DVB ($n = 5$) and AVL action potentials ($n = 17$) in wild-type animals. The error bar is SEM. **d, e** Representative calcium imaging traces from the cell bodies of DVB and AVL expressing GCaMP6f. Left: 300 s continuous recording traces. Right: isolated single calcium spikes from the left traces to demonstrate the difference in calcium signal decay. Source data are provided as a Source Data file.

positive spike in AVL evolved into a much more prominent and regenerative negative spike in the hyperpolarization direction (Fig. 1b, c).

The action potentials in AVL and DVB are likely mediated by voltage-gated calcium currents since the *C. elegans* genome lacks genes encoding voltage-gated sodium channels[26]. In this case, the large and stereotyped calcium-mediated action potential spikes should be detectable using fluorescent calcium indicators expressed in AVL and DVB. Indeed, in immobilized worms expressing GCaMP6f in GABAergic neurons including AVL and DVB, spontaneous and periodic calcium spikes were detected in the cell body of both neurons (Fig. 1d, e). The period of the rhythmic calcium spikes in both neurons are in line with the ~45 s cycle time of the defecation motor program, consistent with previous studies showing periodic calcium transients in AVL and

DVB using different calcium indicators, imaging methods, and genetic backgrounds[9,11,13]. Despite the apparent decay of the peak amplitudes over time due to GCaMP bleaching, the uniform shape and discrete nature of these calcium spikes suggest that they resulted from all-or-none, calcium-mediated action potentials. Presumably due to the presence of the negative spikes terminating calcium signals in AVL, individual calcium spikes in AVL had a shorter spike width than those of DVB (Fig. 1d, e). These results suggest that in addition to the sensory neuron AWA, some motor neurons in the *C. elegans* enteric nervous system also fire calcium-mediated action potentials.

**The AVL action potential is mediated by CaV2 voltage-gated calcium channel UNC-2.** To gain insight into the unusual compound action potential in AVL with an upward spike

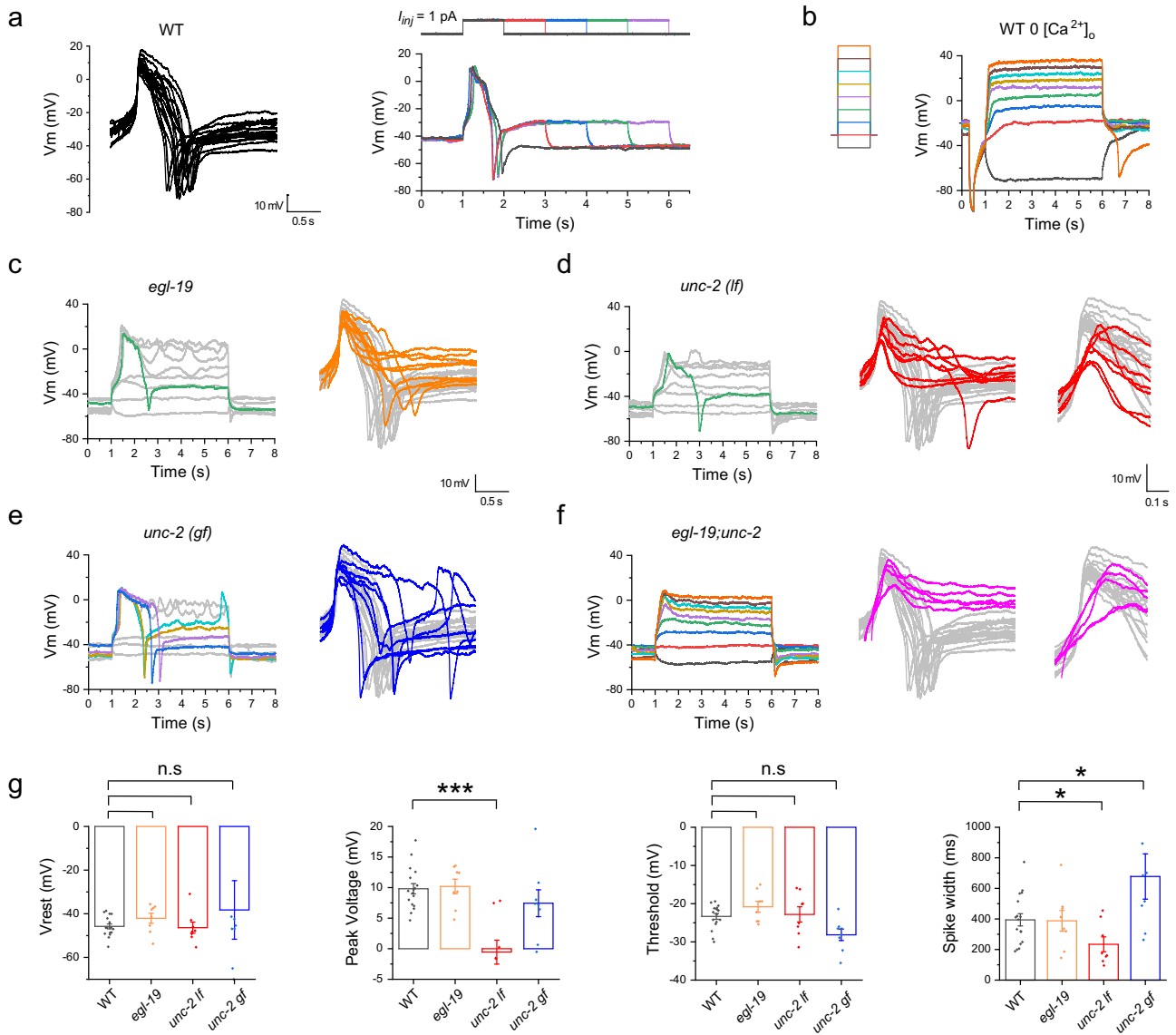

**Fig. 2 AVL action potential is mediated by CaV2 voltage-gated calcium channel UNC-2. a** Left: overlay of isolated single-action potential spikes recorded from 17 animals aligned by the upstroke. Right: the same AVL fires nearly identical action potentials under stimuli with various durations. **b** Positive action potentials in AVL were abolished in calcium-free extracellular solutions but a negative spike was still detected following membrane potential repolarization (orange trace). Left: current-injection stimulation steps (5 s in 1 pA increment). **c–f** Representative current-clamp recordings from AVL in different calcium channel mutants. Left: recording traces under current-injection series as shown in **b**. Right: overlay of single-action potential spikes from different mutants (colored traces) aligned to wild-type traces (same as in **a** but in gray). Action potentials are relatively normal in *egl-19* (**c**), compromised in *unc-2(lf)* (**d**), enhanced in *unc-2(gf)*, and severely compromised in *egl-19;unc-2* (**f**). **d**, **f** Right inserts: zoomed-in view of the depolarization phase showing shallower upstroke in *unc-2 (lf)* and *egl-19;unc-2*. Scale bar is for the right inserts. **g** Statistical comparison of action potential measurements between different genotypes. The averaged peak amplitude and spike width in *unc-2(lf)* are significantly smaller than those of WT. Asterisks indicate significant differences: ***$p = 0.000007$ and *$p = 0.02609$, respectively; the averaged spike width in *unc-2(gf)* is significantly larger than that of the WT (*$p = 0.02172$). Resting membrane potential (Vrest): WT: $-45.7 \pm 1.1$ mV; *egl-19*: $-42.1 \pm 2.3$ mV; *unc-2(lf)*: $-46.4 \pm 2.5$ mV; *unc-2(gf)*: $-38.2 \pm 13.4$ mV. Peak amplitude: WT: $9.8 \pm 0.8$ mV; *egl-19*: $10.2 \pm 1.2$ mV; *unc-2(lf)*: $-0.6 \pm 1.9$ mV; *unc-2(gf)*: $7.4 \pm 2.2$ mV. Threshold: WT: $-23.3 \pm 0.8$ mV; *egl-19*: $-20.8 \pm 1.4$ mV; *unc-2(lf)*: $-22.8 \pm 2.0$ mV; *unc-2(gf)*: $-28.1 \pm 1.5$ mV. Spike width: WT: $394.2 \pm 40.0$ ms; *egl-19*: $387.9 \pm 67.6$ ms; *unc-2(lf)*: $234.3 \pm 48.4$ ms; *unc-2(gf)*: $677.7 \pm 147.8$ ms. The number of animals recorded per genotype: WT: $n = 17$; *egl-19*: $n = 8$; *unc-2(lf)*: $n = 8$; *unc-2(gf)*: $n = 8$. Error bar is SEM. Statistics used was a two-sided two-sample $t$-test. Source data are provided as a Source Data file.

followed by a downward or negative spike, we performed ion substitution and mutant analysis to determine its ionic origin and molecular mechanism. AVL action potentials recorded from wild-type animals in standard recording solutions containing 2 mM $[Ca^{2+}]_o$ had highly stereotyped waveforms under supra-threshold stimuli of various amplitudes and duration (Fig. 2a). When $Ca^{2+}$ was removed from the extracellular solution, the depolarizing spikes in AVL were abolished, consistent with the

hypothesis that the depolarizing spikes in AVL are calcium-mediated action potentials (Fig. 2b). Interestingly, the negative spikes were still detected following membrane repolarization, suggesting that they were generated by a calcium-independent mechanism, which was later determined to be regenerative potassium efflux (see below).

There are three predicted voltage-gated calcium channel (VGCC) family in *C. elegans*[27] and all three are expressed in

AVL[28]. To identify the calcium channels underlying the calcium-mediated positive spike in AVL, we recorded from animals with mutations in the alpha subunit of each VGCC family. In a reduction-of-function mutant of *egl-19*, which encodes a CaV1 L-type VGCC, complete action potentials (with a positive spike followed by a negative spike) were recorded from AVL, although in about half of the cases negative spikes were not reliably evoked (Fig. 2c). In two different loss-of-function (*lf*) mutant alleles of CaV2 P/Q-type VGCC encoded by *unc-2*, complete action potentials with negative spikes are rarely evoked (Fig. 2d and S1d). Moreover, the shape of the positive phase of the AVL spikes were aberrant in *unc-2* mutant compared to the wild type with a shallower upstroke (Fig. 2d) and smaller amplitude (Fig. 2g). The compromised calcium spikes were fully rescued by injecting UNC-2 cDNA under *Punc-47* promotor in one of the *unc-2(lf)* alleles (Fig. S1e). Conversely, in a gain-of-function (*gf*) mutant of *unc-2*[29], complete action potentials were reliably evoked in all recordings, while the duration of AVL action potentials became more variable and significantly longer on average (Fig. 2e, g). These data suggest that UNC-2 plays a primary role in mediating the positive action potential in AVL, possibly with minor contributions from EGL-19. Consistent with this model, in *egl-19;unc-2* double mutant animals, complete action potentials were never evoked in AVL by any stimuli (Fig. 2f). Importantly, periodic expulsion events were also completely eliminated in freely-moving *egl-19;unc-2* double-mutant animals (Fig. S1h), suggesting that calcium-mediated action potential firing in AVL (perhaps also DVB) are crucial in generating expulsion behaviors.

To further examine if other VGCCs play significant roles in generating AVL action potentials, we recorded from mutant animals lacking *cca-1*, which encodes a CaV3 T-type VGCC[30], a double mutant of *egl-19;cca-1*, as well as double mutants of *nca-1;nca-2*, which encode the homologs of the calcium-permeable NALCN channel[31]. We found that AVL action potentials recorded in these mutants were largely unchanged compared to those in the wild type (Fig. S1f, g). However, the frequency of action potentials in *cca-1* and *nca-1;nca-2* mutants appear to be increased (Fig. S1f, g). This casual observation, if holds true, might be explained by a decrease in background cytosol calcium levels, which likely inhibits action potential firing by calcium-dependent UNC-2 channel inactivation.

**The AVL negative spike is mediated by a repolarization-activated potassium channel EXP-2.** To determine the underlying ionic currents that generate the negative spike in AVL, we recorded whole-cell currents in AVL under a voltage-clamp configuration. The most prominent whole-cell currents in wild-type AVL were transient outward currents with fast activating and inactivating kinetics evoked by depolarizing voltage-clamp steps (Fig. 3a). More interestingly, however, was that some small, albeit substantial, outward currents were detected after the cessation of voltage-clamp steps (Fig. 3a, WT insert). Analysis of the current-voltage (I-V) relationship of these small currents indicates that they are voltage-dependent and are activated when the membrane potential is repolarized from above −20 mV back to the holding potential of −60 mV (Fig. 3b). These outward repolarization-activated currents could theoretically generate regenerative spikes in the negative direction by forming a positive-feedback loop (Fig. 3c). The absence of AVL negative spikes in *unc-2(lf)* and *egl-19;unc-2* mutants supports this model, as the generation of negative spikes required strong prior depolarization (Fig. 2d, f).

*exp-2* encodes the only known repolarization-activated potassium channel in *C. elegans*[32–35] and is expressed in AVL[28]. To determine if EXP-2 underlies the repolarization-activated

currents in AVL, we recorded from AVL in *exp-2(lf)* mutant and found that these currents were completely eliminated (Fig. 3a, b). Conversely, in a heterozygous *exp-2(gf)* mutant (homozygous *exp-2(gf)* is lethal), the IV curve of these repolarization-activated currents was left-shifted so that EXP-2(gf)-mediated currents were activated at lower membrane potentials (Fig. 3a, b). Consistent with the prediction that these potassium currents underlie the negative spike in AVL, action potentials recorded in *exp-2(lf)* under current-clamp completely lacked negative spikes while the positive calcium spikes were intact compared to wild-type AVL (Fig. 3d). In contrast, action potentials recorded in *exp-2(gf)* AVL became significantly narrower in spike width, likely resulting from early onset of the negative spike (Fig. 3e, f).

To further test if the effect of EXP-2 on AVL negative spike is cell-autonomous, we rescued *exp-2(lf)* mutant in AVL, DVB and other GABA neurons by injecting *Punc-47* promotor driven *exp-2* cDNA into *exp-2(lf)* animals. We found that the negative spike can be partially restored in *exp-2* rescued animals identified by positive co-injection markers (Fig. S2c). Interestingly, the amplitudes of both calcium spikes and negative potassium spikes in the *exp-2* rescued animals were reduced, which may result from increased leaking currents due to EXP-2 overexpression and/or mis-localization in AVL.

To examine if the transient depolarization-induced currents in AVL that are intact in the *exp-2(lf)* mutant (Fig. 3a) also affect action potential generation, we further identified its ionic origin and tested its potential effect on action potential dynamics. The fast activating and inactivating kinetics of these transient currents are reminiscent of the SHL-1 currents isolated from AWA[14] or reconstituted in *Xenopus* oocytes[36]. Indeed, in *shl-1(lf)* mutant, these transient currents were eliminated under voltage-clamp while the EXP-2-mediated currents remained intact (Fig. S2a). Under the current-clamp, action potentials including both positive and negative spikes were not affected in *shl-1* (Fig. S2d). Consistent with this result, in *exp-2;shl-1* double mutant, in which practically all outward whole-cell currents were eliminated under voltage-clamp, the evoked action potential dynamics under current-clamp were indistinguishable from that of *exp-2*, i.e., normal positive spikes without negative spikes (Fig. S2d). These data suggest that SHL-1 does not play a significant role in generating action potentials in AVL.

**Modeling of the AVL action potential.** To ask whether the EXP-2 current is sufficient to generate the negative spikes, we developed a Hodgkin–Huxley type model of the AVL action potential (See Supplementary Method for the detailed model description). Our AVL model incorporated the various inferred calcium and potassium channels: UNC-2, EGL-19, CCA-1, SHL-1, EXP-2, NCA, and EGL-36 (Kv3) (a voltage-gated potassium channel). The model reproduced many of the qualitative features of the AVL membrane potential dynamics. The voltage-clamp simulations show a prominent transient outward current similar to the experimental data. The small voltage-dependent outward currents after the termination of the voltage-clamp step seen in wild-type AVL were also qualitatively reproduced by the model (Fig. 4a, b). Under current-clamp simulation, there is an excellent agreement in the shape of AVL action potential between the model and the experimental data (Figs. 3c, 4c).

In order to illustrate the role of individual ion channels during an action potential, we further examined the dynamics of each channel in the model (Fig. S3a displays the temporal evolution of the normalized conductance of each modeled channel in wild-type AVL). The stimulating current was set to 10 pA to reliably evoke an action potential. After the current-injection, the CCA-1 inward calcium current was activated first followed by the

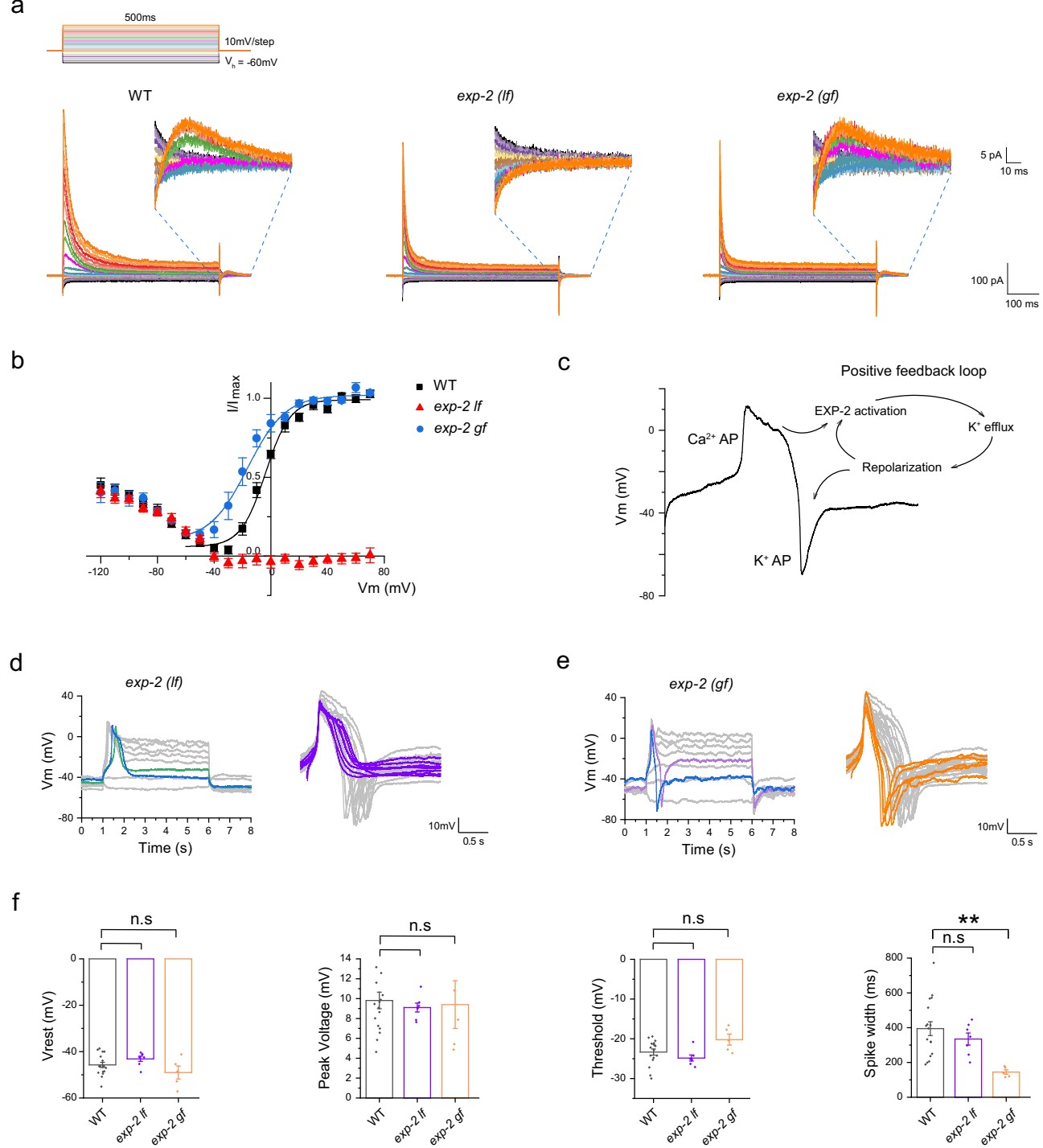

activation of UNC-2 and EGL-19 calcium currents (Fig. S3a). Meanwhile, SHL-1 generated a transient outward potassium current to counter the calcium currents and to limit the peak voltage of the positive spike. These currents contributed to the depolarization phase of the AVL action potential. During the voltage plateau phase of the model, EGL-19 generated a slowly inactivating current, as previously observed experimentally in *C. elegans* pharynx muscles[35]. The UNC-2 calcium currents and the EGL-36 outward potassium current also contributed to the maintenance of the plateau phase (Fig. S3). Finally, when the AVL membrane potential repolarized across a threshold, the

EXP-2 channel activated, rapidly, thus generating an outward current to evoke the negative spike.

To further elucidate the role of EXP-2, we modeled *exp-2(gf)* and *exp-2(lf)* by modifying the parameters used in the AVL model (see Supplementary Methods for detailed description). Consistent with the experimental data (Fig. 3a–e), the I-V curve of the repolarization-activated currents in the *exp-2(gf)* simulation was left-shifted (Fig. 4b) and the currents in *exp-2(lf)* simulation were almost eliminated (Fig. 4b, c). The current-clamp simulations showed AP shapes in *exp-2(gf)* and *exp-2(lf)* that matched the electrophysiological data: a narrower spike width in *exp-2(gf)*, and

**Fig. 3 AVL negative spike is mediated by a repolarization-activated K channel EXP-2. a** Representative whole-cell currents recorded from AVL in wild-type (left panel), *exp-2*(*lf*) (middle panel), and *exp-2*(*gf*) mutant animals (right panel). Top: voltage-clamp steps used: holding potential is −60 mV, each step is 500 ms long in 10 mV increment. Inserts: zoomed-in view of currents from 0 to ~70 ms after returning to holding potential from serial voltage-clamp steps. **b** Current-voltage (I–V) relationships of normalized repolarization-activated currents for WT (black square), *exp-2*(*lf*) (red triangle), and *exp-2*(*gf*) (blue circle). WT and *exp-2*(*gf*) IV curves were fitted with the Boltzmann function (color-matching curves). **c** Schematic of action potential generation in AVL. The negative potassium spike is intrinsically coupled to the preceding positive calcium spike via a positive-feedback loop mediated by repolarization-activated potassium channel EXP-2. **d**, **e** Representative action potential traces recorded from *exp-2*(*lf*) and *exp-2*(*gf*) AVL. Right panels: overlay of isolated single-action potential spikes from *exp-2*(*lf*) (purple traces, note the absence of negative spikes) and *exp-2*(*gf*) (orange traces, note the narrow spike width) aligned to the wild-type traces (gray). **f** Statistical comparison of action potential measurements between different genotypes. The averaged spike width in *exp-2*(*gf*) is significantly narrower than that of the WT. Asterisks indicate significant differences: **$p = 0.00348$. WT measurements are the same as in Figs. 1c and 2g. Resting membrane potential (Vrest): *exp-2*(*lf*): −43.1 ± 1.1 mV; *exp-2*(*gf*): −49.0 ± 2.8 mV. Peak amplitude: *exp-2*(*lf*): 9.1 ± 0.4 mV; *exp-2*(*gf*): 9.4 ± 2.4 mV. Threshold: *exp-2*(*lf*): −24.8 ± 0.8 mV; *exp-2*(*gf*): −20.2 ± 1.4 mV. Spike width: *exp-2*(*lf*): 334.7 ± 34.5 ms; *exp-2*(*gf*): 144.9 ± 12.8 ms. The number of animals recorded per genotype: *exp-2*(*lf*): $n = 7$; *exp-2*(*gf*): $n = 5$. The error bar is SEM. Statistics used was a two-sided two-sample *t*-test. Source data are provided as a Source Data file.

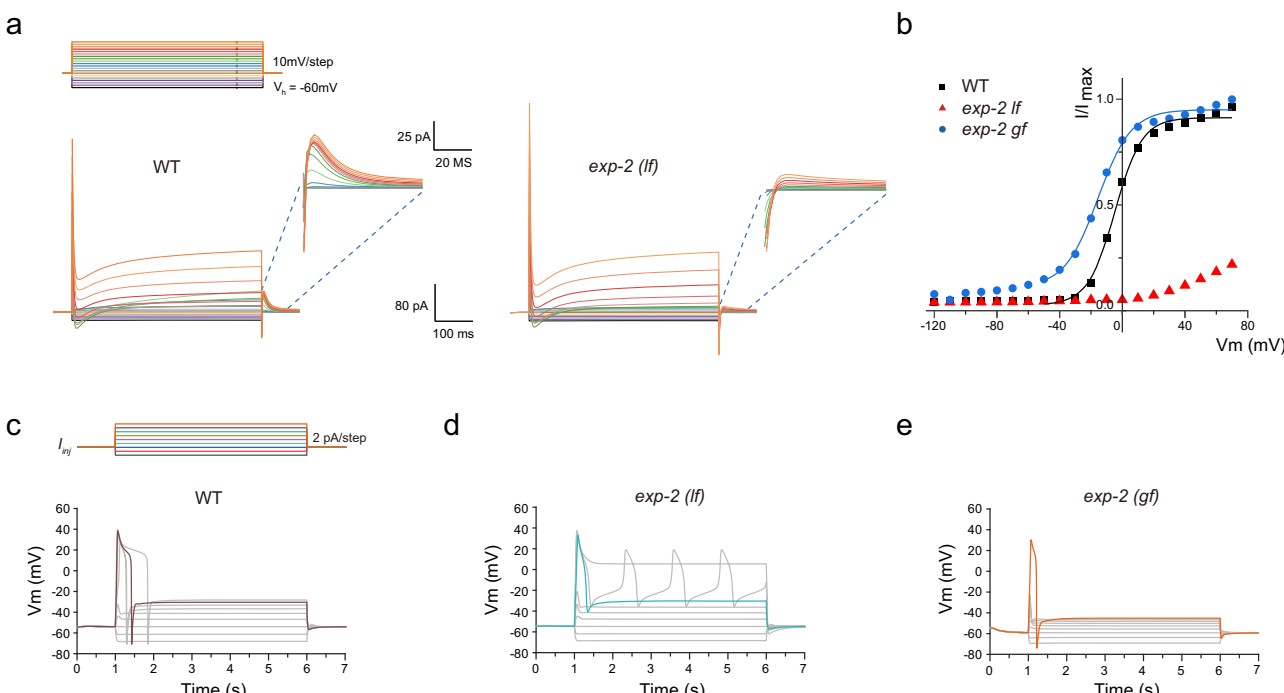

**Fig. 4 Modeling of AVL action potentials. a** Simulation of voltage-clamp of AVL in the wild type (left panel) and the *exp-2*(*lf*) mutant (right panel). Top: voltage-clamp steps used in numerical simulation: holding potential is −60 mV, each step is 500 ms long in 10 mV increment. Inserts: zoomed-in view of currents from 0 to ~60 ms after returning to holding potential from serial voltage-clamp steps. **b** Current-voltage (I–V) relationships of normalized simulated repolarization-activated currents for WT (black squares), *exp-2*(*lf*) (red triangles), and *exp-2*(*gf*) (blue circles). WT and *exp-2*(*gf*) IV curves were fitted with Boltzmann functions (color-matching curves). **c** Representative action potential modeling simulations for wild-type AVL. Top: current-injection steps used in numerical simulation: each step is 5 s long in 2 pA increment. **d**, **e** Representative action potential simulations for *exp-2*(*lf*) and *exp-2*(*gf*) mutants under the same series of current-injection steps to wild-type AVL.

the elimination of the negative spike in *exp-2*(*lf*). These results support our model that the repolarization-activated channel EXP-2 generates the potassium-mediated negative action potential in AVL. We further simulated the effect of increased leaking current in our biophysical model and were able to reproduce a comparable reduction of spike amplitudes in both directions (Fig. S2e, compared to Fig. S2c). These results suggest that the ratio of UNC-2/EXP-2 and/or the subcellular localization of EXP-2 relative to UNC-2 is important for the coupling of positive $Ca^{2+}$ spike and negative $K^+$ spike.

In addition, we also modeled *shl-1* mutant and *exp-2;shl-1* double mutant to examine the role of SHL-1 potassium channel. The simulation results agree with the experimental data in Fig. S2c. The current-clamp simulation of *shl-1* mutant showed a similar AP shape to wild-type AVL (Fig. S3d); and the simulation of *exp-2;shl-1* double mutant showed a similar AP shape to *exp-2*(*lf*), with the

absence of the negative spike (Fig. S3e). The combined experimental and modeling results suggest that SHL-1 can generate a transient inward potassium current to counteract the calcium currents in the upstroke phase, but does not play a significant role in other phases.

**AVL and DVB synchronously fire propagating action potentials.** What is the physiological and functional significance of enteric neurons firing action potentials in defecation behavior? What is the timing of action potentials in AVL and DVB relative to each other and to the intestinal calcium clock? How does spike timing affect the downstream expulsion behavior? To address these questions, we expressed GCaMP6f in AVL and DVB as well as in the intestine and recorded calcium dynamics in the enteric system in freely-moving animals. We used a closed-loop live tracking microscope[37] to monitor the worm's body length to register body muscle contraction and expulsion events and

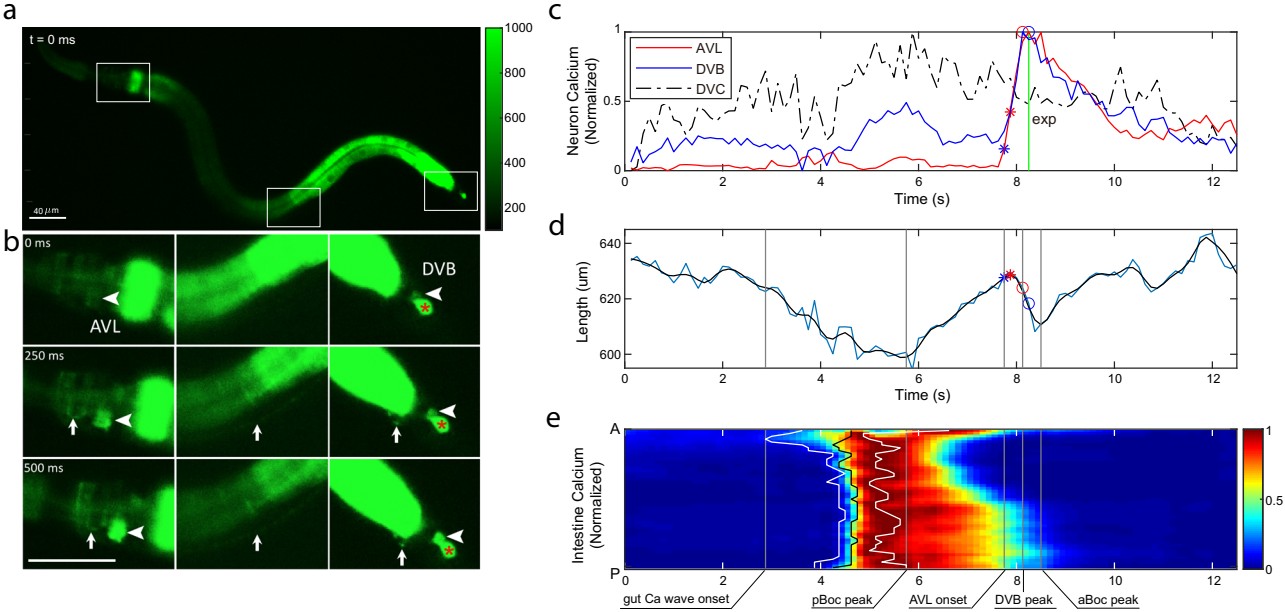

**Fig. 5 AVL and DVB synchronously fire propagating action potentials. a, b** Representative images showing synchronous GCaMP fluorescence in AVL and DVB. Time 0 is an arbitrary frame before neural activation. AVL soma (left panel, arrowhead), AVL axon (left and middle panels, arrow), preanal NMJs (right panel, arrow), DVB soma (right panel, arrowhead). Same scale bar for **a, b**. This imaging strain also has GCaMP expressed in DVC (marked by a red * in the right panel) which is irrelevant to DMP but used as a negative control. 13 imaging experiments were done with similar results. **c** Representative (normalized) calcium activities in AVL (red), DVB (blue), and DVC (black) soma. All activities normalized with maximum set to 1. Vertical line and markers: The timing of expulsion (green vertical line), the onset (asterisk), and the peak (circle) of AVL and DVB spikes. **d** Corresponding worm body length as a function of time. The blue curve is frame-by-frame and black is a smoothed version of the blue curve. (Smoothing used local regression to a second-order polynomial.) Red and blue asterisk/circle: same as in **c**. **e** Corresponding (normalized) intestinal calcium activities along the worm body axis (A: anterior, P: posterior). Curves (from left to right): the timing of the onset (white), the half-peak (black), and the peak (white) of calcium activities along the worm body axis. Vertical lines in **d, e** (from left to right): the earliest onset of intestinal calcium activity, the peak of pBoc, the onset of AVL, the peak of DVB, and the peak of aBoc. Source data are provided as a Source Data file.

correlated these motor steps to simultaneously recorded calcium dynamics in the gut-neuron circuit. Since AVL and DVB are positioned roughly at the same z-axis level, cell bodies from both neurons can be imaged in the same focal plane. In wild-type animals, rhythmic intestinal calcium waves, calcium spikes in AVL and DVB indicative of action potentials, and body muscle contraction steps were reliably recorded (Fig. 5, S4, and Movie 1). The peak of posterior body muscle contractions preceded the peak of anterior body muscle contractions by ~2.5 s and the start of the intestinal calcium wave by ~1 s (Fig. S4c); the averaged cycle length was ~45 s with small variability (Fig. S4c, d). Synchronized calcium spikes in AVL and DVB coincided with anterior body muscle contractions and expulsions but lagged about 3 s behind the intestinal calcium wave (Fig. 5c–e). These measurements are consistent with the existing model that the posterior body muscle contraction is directly activated by calcium-mediated proton release from the intestine[8] and that AVL and DVB drive anterior body muscle contraction and expulsion steps downstream of the intestinal clock[2–4,6,7,9,11]. Most interestingly, under ideal imaging conditions, we detected synchronous calcium spikes in the entire AVL axon together with calcium spikes in AVL and DVB cell bodies (Fig. 5a, b), consistent with a previous study showing AVL to be isotonically activated[11]. As a negative control, calcium signals in DVC did not show time-locked changes with intestinal calcium wave, AVL/DVB calcium spike, or defecation motor steps (Fig. 5b, c). These observations suggest that action potentials propagate along the AVL axon which likely underlies the synchronous firing of AVL and DVB.

**Synchronous firing of AVL and DVB underlies reliable expulsion**. To further improve imaging and tracking quality and facilitate data analysis for longer recording covering multiple

defecation cycles, we constructed a new imaging line with co-expression of mCherry and GCaMP in the enteric nervous system and the intestine (Fig. 6a) and employed a two-color imaging procedure to simultaneously acquire both GCaMP dynamics and mCherry fluorescence signals (Movie 2). Calcium-independent mCherry fluorescence was used as a reference for localizing the cell body and minimizing motion artifacts[37]. To objectively identify the three motor steps of the defecation cycle, we analyze the body contraction by dividing the worm body into two halves and measuring each half of the body length separately (Fig. S5a). Posterior body muscle contractions are strong and thus were clearly detected in all cycles coinciding with intestinal calcium waves (Fig. 6b and S5b). From the computed body lengths alone, sometimes the smaller anterior body muscle contraction and expulsion events were indistinguishable from noise; however, even in most of these cases, these two steps could be easily identified by eye. When both anterior body muscle contraction and expulsion events were clearly visible in the same cycle, we found that they were always initiated at roughly the same time (Fig. 6b and S5b).

Using the dual-color strain and imaging setup in the wild-type genetic background, we found that calcium spikes fired in AVL almost always slightly preceded DVB spikes with an average temporal advance of 215 ± 176 ms (Fig. 7d). In some cases, AVL and DVB spikes were nearly overlapping with very small delays between them, but DVB spikes were not detected before AVL spikes (Fig. 7d). This result and the observation of propagating calcium spikes along the AVL axon (Fig. 5b) suggest that spike signals flow from AVL to DVB.

In a recent study, Choi et al. reported that AVL and DVB activities are electrically coupled by the gap junction INX-1

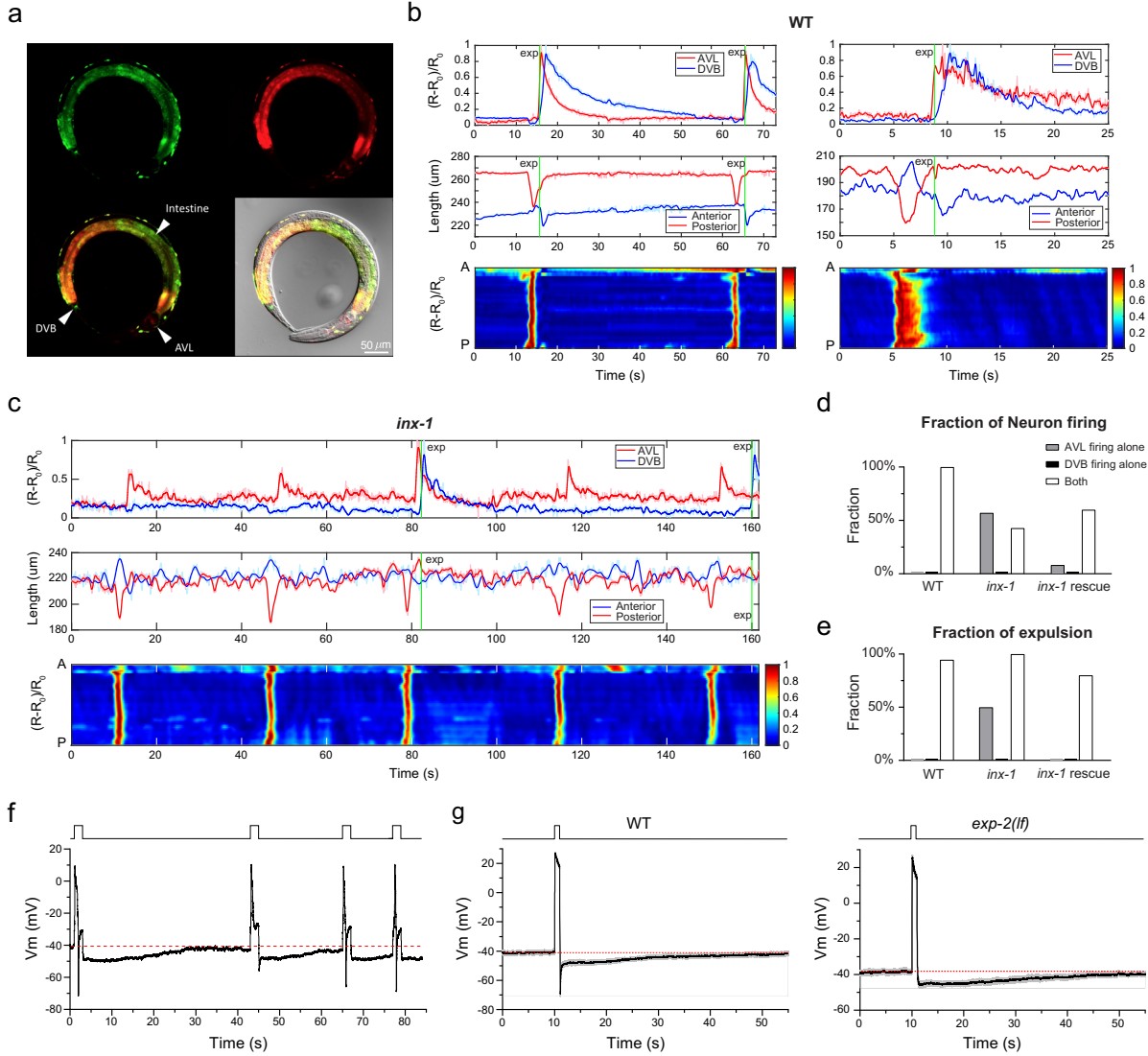

**Fig. 6 Synchronous action potential firing in AVL and DVB underlies reliable expulsion. a** Representative fluorescent and DIC images of the dual-color imaging line used for Figs. 6, 7. An anesthetized L4 worm with co-expression of mCherry (red) and GCaMP6f (green) in all GABA neurons and the intestine. Seven randomly selected animals were imagined with near-identical results as the example shown here. **b**, **c** Top panels: Representative calcium activities of AVL (red) and DVB soma (blue) in wild-type (**b**) and *inx-1* mutant animals (**c**). Middle panels: Corresponding traces showing simultaneous measurements of the length of the anterior (blue) and posterior (red) half of the centerline. Bottom panels: Corresponding normalized intestinal calcium activities along the worm body axis (A: anterior, P: posterior). Green vertical lines: Timing of expulsion. **d** Fraction of AVL and DVB firing detected in DMP cycles (WT: *n* = 19 from 12 animals; *inx-1*: *n* = 14 from 3 animals; *inx-1*(rescue): *n* = 25 from 5 animals). **e** Fraction of expulsion detected under the circumstances where only AVL fires, only DVB fires, or both neurons fire (WT: *n* = 19 from 12 animals; *inx-1*: *n* = 14 from 3 animals; *inx-1*(rescue): *n* = 25 from 5 animals). In **d**, **e**, the values for AVL or DVB firing alone in WT and DVB firing alone in *inx-1* are zero. **f** A representative current-clamp recording trace of AVL under repetitive stimuli (each stimulus is 3-s long at 2 pA) at various intervals. **g** The averaged trace of prolonged current-clamp recording of AVL under a 1-s long 8 pA stimulus in WT (left panel) and *exp-2*(*lf*) mutant (right panel). Gray: SEM, *n* = 7. Red dotted lines in **f**, **g** label the resting membrane potential. Source data are provided as a Source Data file.

located at the preanal ganglia NMJs[11]. Consistent with this finding, we found that calcium spikes in AVL and DVB were de-synchronized in *inx-1*(*lf*) mutant background (Fig. 6c and S5c), and DVB spikes were only detected in 43% of cycles in *inx-1*(*lf*) compared to 100% in WT (Fig. 6d). In cycles in which both AVL and DVB fired, the delay time between the AVL and DVB spikes was significantly increased in *inx-1* (2847 ± 2307 ms) (Fig. 7d). DVB firing spikes alone without AVL firing was never observed in either WT (*n* = 17 cycles) or *inx-1* (*n* = 14 cycles) (Fig. 6d). Furthermore, most firing and expulsion defects in *inx-1*(*lf*) (e.g., AVL firing alone, asynchronized DVB firing, and ectopic expulsions along with DVB firing) were fully rescued by

expressing INX-1cDNA in AVL and DVB under *Punc-47* promotor (Figs. 6d, e, 7c, d and Fig. S5d). However, the reduced fractions of cycles with synchronized AVL/DVB firing and detectable expulsions were only partially or not rescued (Figs. 6d, 7b). Whether this is due to overexpression of INX-1 in other GABA neurons remains to be determined. Nevertheless, our results and a previous study by Choi et al. collectively support a model in which AVL fires action potentials first, which propagate orthodromically to the preanal ganglia and activate DVB to fire action potentials through the INX-1 gap junction.

Our behavioral tracking and simultaneous recording of calcium spikes in AVL and DVB also support previous studies

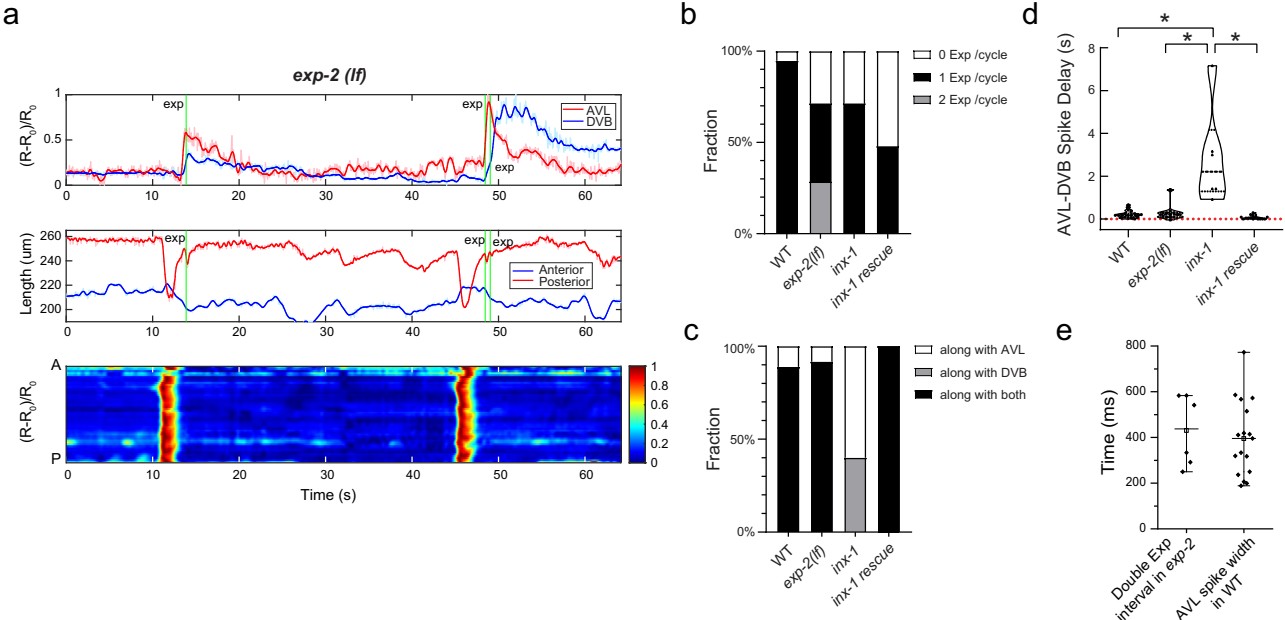

**Fig. 7 The AVL negative spike inhibits excessive expulsion per defecation cycle. a** Top: Representative (normalized) calcium activities of AVL (red) and DVB soma (blue) in *exp-2(lf)* mutants. Middle: Corresponding changes of the length of the anterior (blue) and posterior (red) half of the worm body. Bottom: Corresponding normalized intestinal calcium activities along the worm body axis (A: anterior, P: posterior). Green vertical lines: Timing of expulsion. **b** Fraction of DMP cycles associated with different numbers of expulsion occurrence (WT: *n* = 19 from 12 animals; *exp-2(lf)*: *n* = 21 from 6 animals; *inx-1*: *n* = 14 from 3 animals; *inx-1*(rescue): *n* = 25 from 5 animals). **c** Fraction of expulsions per cycle associated with AVL or DVB firing or both (WT: *n* = 18 from 12 animals; *exp-2(lf)*: *n* = 12 from 6 animals; *inx-1*: *n* = 10 from 3 animals; *inx-1*(rescue): *n* = 12 from 5 animals). **d** Time delay between AVL and DVB spikes in wild-type, *exp-2(lf)*, *inx-1*, and *inx-1* rescued animals (WT: *n* = 19 from 12 animals; *exp-2(lf)*: *n* = 12 from 6 animals; *inx-1*: *n* = 6 from 3 animals; *inx-1*(rescue): *n* = 12 from 5 animals). Data were presented in a violin plot. Asterisks indicate significant differences: *\*P* < 0.05 in unpaired Student's *t*-test with Welch's correction. The statistical test was two-sided. WT vs *inx-1*: *P* = 0.0382; *inx-1* vs *inx-1*(rescue): *P* = 0.0318; *exp(lf)* vs *inx-1*: *P* = 0.0413. **e** Time delay between two consecutive expulsions within the same cycle in *exp-2(lf)* mutants (*n* = 6) and the spike width of AVL action potentials in wild-type animals (*n* = 17). Data were presented as mean values ± SEM. The center is the mean value and the error bar is SEM. Source data are provided as a Source Data file.

suggesting that both enteric neurons are semi-redundantly required for reliable expulsion[4,11]. In wild-type animals, expulsion was reliably detected in 94.7% cycles (18 out of 19 cycles) and was always associated with synchronized AVL/DVB spikes (Fig. 6e). In the *inx-1* mutant, when AVL fired alone in a cycle, expulsions were only observed in 50% of cycles (*n* = 8) (Fig. 6e), while when both AVL and DVB fired, even with longer delays, expulsions were detected in 100% of cycles (Fig. 6e). However, the exact timing of expulsions in the *inx-1* mutant was more variable: expulsion coinciding with the AVL spike in 33% of cycles, and with the DVB spike in 67% of cycles (*n* = 6) (Fig. 7c). These data confirm that reliable expulsion behavior required activation of both AVL and DVB[4,11], and also suggest that the timing of expulsion is encoded by the synchronicity of the spike code from two spatially well-separated but electrically-coupled enteric neurons.

**AVL action potentials are followed by long-lasting after-hyperpolarization.** Our observation of asynchronous DVB spikes only in the *inx-1* mutant but not in the wild type supports a recently proposed model of INX-1 gap junction suppressing ectopic DVB spiking during intercycles[11]. To explore the electrophysiological mechanism of this hypothesis, we recorded AVL for prolonged periods of time after inducing action potentials (Fig. 6f). Strikingly, we found that in addition to the negative spike, there was a long-lasting AHP following each action potential (Fig. 6f). We accurately measured the length of the long-lasting AHP by inducing an action potential like waveform in

AVL using a 1-s-long stimulus (Fig. 6g, left) and found the averaged AHP lasted for ~40 s, matching the duration of the intercycle. This long AHP is independent of the negative spike, as it was present in *exp-2(lf)* mutant (Fig. 6g, right). Since INX-1 forms bidirectional homotypic gap junctions between AVL and DVB[11], this slow hyperpolarization might pass through the gap junction and keep DVB inactive during intercycles in wild-type animals. In the *inx-1* mutant, elimination of this inhibitory entrainment force by the removal of the gap junction could allow DVB to fire at its own pace, manifested as asynchronized firing of DVB with variable delays after firing of AVL (Fig. 7d).

**The AVL negative spike inhibits excessive expulsion per defecation cycle.** To determine the physiological function of the AVL negative spike during defecation, we performed live imaging and behavioral tracking in the *exp-2(lf)* mutant in which AVL fire normal calcium action potentials without negative spikes (Fig. 3d). We found that *exp-2(lf)* did not affect the cycle period nor the delay between AVL and DVB spikes (Fig. 7d); however, in ~30% of total cycles, an additional expulsion was observed immediately following the normally timed expulsion (Fig. 7a, b and S6). The average delay between the two consecutive expulsions in *exp-2(lf)* was 430 ± 155 ms (*n* = 6), which corresponds well with the averaged spike width of AVL action potential (394.2 ± 40.0 ms, *n* = 17) (Fig. 7e). This result suggests that negative spikes in AVL may prevent excessive expulsions by terminating calcium signals in AVL and/or DVB in a timely manner. Furthermore, in *exp-2(lf)* only ~40% of cycles were

associated with a normal single expulsion, and in ~30% of cycles no expulsions were detected at all (Fig. 7b), suggesting that the AVL negative spike may play a more important role in regulating the enteric rhythm and coordinating motor steps during defecation behavior.

## Discussion

Here we identify two new spiking neurons, AVL and DVB, in the *C. elegans* enteric nervous system, with distinct membrane potential dynamics and putative functions. DVB fires giant calcium-mediated action potentials while AVL fires unusual compound action potentials with each giant calcium spike followed by a potassium-mediated negative spike. Electrophysiological recording and biophysical modeling suggest that a CaV2 type voltage-gated calcium channel UNC-2 and a repolarization-activated potassium channel EXP-2 underlie the molecular machinery generating the time-locked compound action potential in AVL. Using simultaneous neuron-gut imaging and behavioral tracking in freely-moving animals, we found that AVL and DVB synchronized action potential firing through electrical coupling by INX-1 gap junction, and that the AVL negative spike prevented excessive expulsions in a given defecation cycle. Our study provides new insight into temporal coding, spike propagation and signal integration within the *C. elegans* enteric neural circuit underlying motor pattern generation and behavioral rhythm.

**The AVL negative spike**. The negative spikes recorded in AVL resemble those observed in the pharyngeal muscles of *C. elegans* and other nematodes[32,38] as well as in lobster muscles[39]. It is worth noting that all these excitable cells firing negative spikes are elements of apparatuses that produce biological rhythms, suggesting that the negative spike in AVL may be involved in generating and/or modulating the enteric rhythm underlying the defecation motor program. Our electrophysiological recording results suggest that the AVL negative spike is generated by EXP-2-mediated repolarization-activated potassium currents. Although we still cannot completely rule out the possibility that a *Punc-47* + neuron forming reciprocal inhibitory connection with AVL may theoretically be responsible (such hypothetical connection cannot be from DVB because no EXP-2 currents were detected in DVB as shown in Fig. S2b) for generating negative spikes, the fact that EXP-2-mediated repolarization-activated $K^+$ currents were recorded in AVL and our mathematical single-cell AVL model could simulate precisely the shape and time course of the negative spike strongly suggest a cell-autonomous effect. From our model voltage dynamics, the AVL negative spike appears to be an upside-down version of the Hodgkin–Huxley action potential (Fig. 3c), with the activation and inactivation of EXP-2 analogous to the activation of Na and the activation of K in the Hodgkin–Huxley model. More specifically, in wild-type AVL, the repolarization of preceding calcium-mediated action potential activates EXP-2, allowing potassium efflux which induces further repolarization and more EXP-2 activation, forming a positive-feedback loop that results in a regenerative negative spike. Thus, the positive calcium spike and negative potassium spike are intrinsically coupled, producing a biophysical mechanism to maintain and regulate the duration of calcium influx during each action potential firing.

To our knowledge, AVL is the only known neuron that fires this characteristic negative spike mediated by a repolarization-activated potassium channel in *C. elegans* or other animals. In addition to AVL, *exp-2* is also expressed in pharyngeal muscles and a few other neurons in *C. elegans*[32]. Presumably, through a similar mechanism, EXP-2 in pharyngeal muscles also functions

in regulating the rhythmic pharyngeal action potentials and the pumping rate[32]. It may be worth looking into other *exp-2* expressing neurons to investigate their potential roles in generating or regulating rhythmic behaviors in *C. elegans*.

The ion channel composition that generates action potentials in DVB was not electrophysiologically examined here; however, previous calcium imaging studies suggest that EGL-19 mediates calcium transient in DVB with a possible minor contribution from additional calcium channels[11,13]. Furthermore, our whole-cell currents recorded in wild-type DVB show very little inward or outward currents under a broad range of voltage-clamp steps (Fig. S2b). This result suggests that the repolarization of DVB action potential does not rely on potassium conductances but rather mainly on voltage-gated calcium channel inactivation.

Our observation of frequent double-expulsion events in the *exp-2(lf)* mutant suggests a plausible physiological function of the AVL negative spike. Excessive expulsions at the wrong time not only waste energy but may also cause more severe detrimental problems to the defecation cycle, e.g., ~30% cycles in *exp-2(lf)* lack expulsion steps (Fig. 7b). This consequence could be induced by disrupting the normal temporal rhythm through unknown retrograde feedback signals from enteric muscles to the enteric nervous system. Interestingly, in the *inx-1* mutant where DVB action potentials are out of alignment, double expulsions are never observed. This suggests that most likely the AVL negative spike functions in limiting calcium signals from AVL itself rather than from DVB through INX-1. Future experiments with *exp-2(lf);inx-1* double mutant would draw more conclusive pictures.

**Timing of anterior body muscle contraction and expulsion**. Almost all ethograms in the literature label the expulsion step behind the anterior body muscle contraction step by one time-unit (usually 1 s)[3,7,40], sometimes with a graphical representation of one full second in between the two steps[41]. Using our customized whole-animal imaging and behavioral tracking setup, we detect nearly concurrent anterior body muscle contraction and expulsion events in freely-moving wild-type animals (Fig. 6b and S5b), which may seem inconsistent with the literature regarding the timing of these two steps. However, manually scoring anterior body muscle contraction for plotting ethograms is known to be not wholly reliable due to the locomotory movements of the head region[3]. The whole-worm tracking of the defecation cycle in the current study is, to the best of our knowledge, the first unbiased quantitative measurement of the timing of these two motor steps in freely-moving animals. The near concurrence of anterior body muscle contraction and expulsion is consistent with their common control by intestinal neuropeptide signaling[3,10] and the instantaneous action potential propagation along the AVL axon from head to tail (Fig. 5b).

**Neural regulation of the defecation behavior**. The identification of action potentials in AVL and DVB highlights a previously underestimated role played by the enteric nervous system in regulating the defecation cycle. In retrospect, at least three features of the defecation motor program demand the all-or-none property and temporal precision that are best supported by spiking neurons. First, the expulsion step requires three muscle groups including the intestinal muscle, sphincter muscle, and anal depressor muscle to contract and relax in synchrony to execute a successful expulsion. The broad calcium-mediated action potentials synchronously fired by AVL and DVB ensure an ample dose of GABA release to orchestrate postsynaptic muscle activities. Second, the defecation motor program requires temporal and spatial coordination of multiple organs from head to tail including the intestine, anterior and posterior body-wall muscles,

two spatially separated enteric neurons, and multiple muscle groups for executing expulsion. The intestinal calcium wave that lasts several seconds without well-defined temporal and spatial initiation and termination time points generates the crude pacemaker signal but is unlikely to provide precision control and coordination across all the downstream players. Rather, near-instantaneous action potential spikes could digitalize intestinal calcium waves and propagate the spike code from AVL to DVB to precisely synchronize the apparatus between the head and tail during each cycle. Third, presumably, all behavioral rhythms and the underlying clocks and pattern generators last the entire lifetime of an animal. This likely requires robust error-correcting and fail-safe mechanisms to maintain stability by preventing the accumulation of temporal fluctuations or even system breakdowns. The rhythmic firing of action potentials, the presence of negative spikes, and prolonged AHPs in many rhythm-generating systems strongly suggest a universal rhythm regulation scheme.

Our work suggests that spiking neurons AVL and DVB with distinct membrane dynamics may play differential roles in the defecation motor program. Specifically, DVB may function mainly as a workhorse to facilitate driving downstream muscle contractions, while AVL may act as a neuronal timekeeper functioning at multiple timescales: At the ultrafast, millisecond time scale, AVL instantaneously relays gut pacemaker signals through propagating action potential along its axon from head to tail to activate DVB across gap junctions; at the fast time scale on the order of seconds, AVL negative spikes function to timely terminate calcium signals and therefore suppress excessive expulsion; at the slow behavioral time scale of the defecation cycle, the long-lasting AHP in AVL suppresses ectopic DVB firing during intercycles by negative entrainment through gap junctions. All in all, this study supports a model in which spiking enteric neurons provide additional timing information at both fast and slow scales to fine-tune and coordinate the defecation motor steps initiated by the intestinal calcium clock.

**INX-1 gap junctions mediate coordinated action potential firing in AVL and DVB.** Recently, a study by Choi et al. demonstrated that the INX-1 gap junction between AVL and DVB functions in the bidirectional electrical coupling of this neuron pair underlying expulsion behavior[11]. Our independent study using different imaging and behavioral tracking setup of animals with different genetic backgrounds supports the prior conclusion and further provides a plausible electrophysiological mechanism. Both studies detected near synchronous calcium spikes in AVL and DVB in wild-type animals (Figs. 5c, 6b and ref. [11]. The instantaneous calcium spike propagation along the AVL axon (Fig. 5b and ref. [11]) and the slight temporal advance of the AVL spike (Figs. 6b, 7d) naturally suggest the following model: NLP-40 release from the anterior intestine induces AVL action potentials first at the cell body or at the axon; AVL propagates the action potential along the axon to preanal NMJs and evokes action potentials in DVB through INX-1 gap junctions. Furthermore, both studies detected ectopic DVB firing in the inx-1 mutant at a slower rate and mis-aligned with respect to the rest of the steps (averaged DVB firing cycle was 84.6 s in this study, n = 3, Fig. 6c and ref. [11]), and expulsions that occur in the absence of INX-1 gap junctions (Fig. 7b, c) or AVL activation[11]. These observations suggest that NLP-40 diffused across the pseudocoelom can activate DVB and induce expulsion, at least in the absence of AVL (and its long-lasting AHP). In wild-type animals, such out-of-phase DVB activation is suppressed by INX-1[11], which can be explained by our observation of the long-lasting AHP in AVL at the time scale of the intercycle (Fig. 6f, g): prolonged AVL AHP hyperpolarizes DVB through INX-1 gap

junctions during intercycles and prevents out-of-phase action potential firing in DVB induced by slow-diffusing NLP-40. Taken together, we propose the following scenario: AVL bidirectionally entrains the enteric nervous system through INX-1 gap junctions, allowing the gut pacemaker signals encoded in the AVL action potentials to be propagated across the whole-body length to DVB; Eventually, the integrated spike code from AVL and DVB determine the probability of an all-or-none behavior output – the expulsion.

**Biophysical diversity of the *C. elegans* nervous system.** To date, three spiking neurons, including a sensory neuron and two motor neurons, have been identified in *C. elegans* (Liu et al., 2018 and this study). This suggests that the *C. elegans* nervous system is likely more heterogeneous than previously thought. Digital neural coding seems to play significant roles in certain computational and behavior functions in *C. elegans*, i.e., encoding sensory stimuli in AWA[14], and long-distance signal propagation and temporal coordination of AVL and DVB coupled by gap junctions. We are just beginning to appreciate the importance of the biophysical diversity and associated functions among different neuronal cell types in the *C. elegans* nervous system. In order to understand circuit function and system-level activity dynamics, the intrinsic biophysical properties of individual neurons need to be thoroughly investigated with electrophysiological recording, optical imaging techniques, as well as theoretical modeling. Equipped with the complete connectome of this small brain, it is conceivable to develop an anatomically correct and biophysically accurate systems model in the future to gain a comprehensive understanding of the *C. elegans* nervous system. Using this approach, this work provides a case study of investigating motor pattern generation and gut-brain communication—two universal biological functions across the animal phyla.

## Methods

*C. elegans* **culture and strains**. All animals used in this study were maintained at room temperature (22–23 °C) on nematode growth medium (NGM) plates seeded with *E. coli* OP50 bacteria as a food source[42]. Wild-type animals were Bristol strain N2 or PD1074.

CX18004 *kyIs772* strain was made by integration (UV irradiation) of CX17967 *kyEx6380*. Extrachromosomal array *kyEx6380* was made by co-injecting N2 strain with 50 ng/ul plasmid pQL1 containing *Punc-47mini::GCaMP6f::unc-54 3′UTR* and 5 ng/ul plasmid pQL2 containing *Pinx-16::GCaMP6f::unc-54 3′UTR*. The unc-47mini promoter (*Punc-47mini*) is 196 bp upstream of ATG of *unc-47*, amplified using the primer pair of aattaaGGCCGGCCggtctaataatcccccgtgctcttc and gagacgacacgcgtcacatttatttcattacagGGCGCGCCaattaa. The *inx-16* promotor (*Pinx-16*) is amplified using the primer pair of aattaaGGCCGGCCaagacaaagtgactcagtg and aatttcagggtacaccaaaGGCGCGCCaattaa. CX18153 *kyIs777* strain was made by integration of the extrachromosomal arrays CX18103 *kyEx6441;kyEx6442* which were crossed from CX18089 *kyEx6441* and CX18090 *kyEx6442*. *kyEx6441* was made by injecting PD1074 strain with 30 ng/ul plasmid pQL6 containing *Punc-47::GCaMP6f::SL2::mCherry::let858 3′UTR*. *kyEx6442* was made by co-injecting PD1074 strain with 50 ng/ul plasmid pQL7 containing *Punc-47mini::GCaMP6f::SL2::mCherry::unc-54 3′UTR* and 5 ng/ul plasmid pQL2 containing *Pinx-16::GCaMP6f::unc-54 3′UTR*. The *unc-47* promoter (*Punc-47*) was amplified using the primer pair of aattaaGGCCGGCCactaaacttctacgtcaaaaagttgacaaaac and gagacgacacgcgtcacatttatttcattacagGGCGCGCCaattaa.

*unc-2* rescue strain CX18364 *kyEx6593* was made by co-injecting 100 ng/ul plasmid pQL13 containing *Punc-47::synthetic intron::pBluescript SK(-)::unc-2 cDNA with synthetic intron::N-GFP::unc-54 UTR* and 10 ng/ul coelomocyte marker plasmid *Punc-122::dsRed* into CX18359 *unc-2(lj1) X; lin-15B(n765) X; oxIs12 X* strain. Worms with red coelomocyte markers were picked for electrophysiological experiments. *exp-2* rescue strain CX18362 *kyEx6591* was made by co-injecting 89 ng/ul plasmid pQL11 containing *Punc-47:: exp-2 cDNA::SL2-mCherry::let858 3′ UTR* and 10 ng/ul *Punc-122::dsRed* into CX17773 *exp-2(sa26ad1426) V; lin-15B(n765) X; oxIs12 X* strain. Worms with red coelomocyte markers were picked for electrophysiological experiments. *inx-1* rescue strain CX18360 *kyEx6589* was made by co-injecting 25 ng/ul plasmid pHW175[11] (a gift from Derek Sieburth) containing *Punc-47::inx-1a cDNA* and 10 ng/ul *Punc-122::dsRed* into CX18176 *kyIs777II; inx-1(tm3524) X* strain. Worms with red coelomocyte markers were picked for imaging experiments.

The complete strain list and plasmid list used in this study are provided in Tables 1 and 2, respectively.

**Electrophysiology**. Electrophysiological recordings were performed as previously described in ref. [14] on young adult hermaphrodites (~3 days old). The gluing and dissection were performed at room temperature under an Olympus SZX16 stereomicroscope equipped with a 1X Plan Apochromat objective and widefield 10X eyepieces. Briefly, an adult animal was immobilized on a Sylgard-coated (Sylgard 184, Dow Corning) glass coverslip in a small drop of DPBS (D8537; Sigma) by applying a cyanoacrylate adhesive (Vetbond tissue adhesive; 3M) along the dorsal side of the head or tail region. A puncture in the cuticle away from the incision site was made to relieve hydrostatic pressure. A small longitudinal incision was then made using a diamond dissecting blade (Type M-DL 72029 L; EMS) along the glue line either between two pharyngeal bulbs for AVL recording or in the tail region for DVB recording. The cuticle flap was folded back and glued to the coverslip with GLUture Topical Adhesive (Abbott Laboratories), exposing the neuron of interest. The coverslip with the dissected preparation was then placed into a custom-made open recording chamber (~1.5 ml volume) and treated with 1 mg/ml collagenase (type IV; Sigma) for ~10 s by hand pipetting. The recording chamber was subsequently perfused with the standard extracellular solution using a custom-made gravity-feed perfusion system for ~10 ml.

All electrophysiological recordings were performed with the bath at room temperature under an upright microscope (Axio Examiner; Carl Zeiss, Inc) equipped with a 40X water immersion lens and 16X eyepieces. AVL and DVB were identified by GFP, mCherry, or GCaMP markers under fluorescent illumination and their physical position. Preparations were then switched to the differential interference contrast (DIC) setting for the patch-clamp. Electrodes with resistance (RE) of 20–30 MΩ were made from borosilicate glass pipettes (BF100-58-10; Sutter Instruments) using a laser pipette puller (P-2000; Sutter Instruments) and fire-polished with a microforge (MF-830; Narishige). We used a motorized micromanipulator (PatchStar Micromanipulator; Scientifica) to control the electrodes backfilled with standard intracellular solution. The standard pipette solution was (all concentrations in mM): [K-gluconate 115; KCl 15; KOH 10; MgCl$_2$ 5; CaCl$_2$ 0.1; Na$_2$ATP 5; NaGTP 0.5; Na-cGMP 0.5; cAMP 0.5; BAPTA 1; Hepes 10; Sucrose 50], with pH adjusted with KOH to 7.2, osmolarity 320–330 mOsm. The standard extracellular solution was: [NaCl 140; NaOH 5; KCL 5; CaCl$_2$ 2; MgCl$_2$ 5; Sucrose 15; Hepes 15; Dextrose 25], with pH adjusted with NaOH to 7.3, osmolarity 330–340 mOsm. For calcium-free extracellular solution, 2 mM CaCl$_2$ was replaced with 2 mM EGTA. Chemicals were from Sigma-Aldrich. Liquid junction potentials were calculated and corrected before recording. Whole-cell current-clamp and voltage-clamp experiments were conducted on an EPC-10 amplifier (EPC-10 USB; Heka) using PatchMaster software (Heka). Two-component capacitive compensation was optimized at rest, and series resistance was compensated by 50%. Analog data were filtered at 2 kHz and digitized at 10 kHz. Current-injection and voltage steps were applied through the recording electrode. For current-clamp recording, a prepulse of −10 pA current-injection was applied before each step to measure the input resistance to make sure that the seal resistance of the patch has not been changed during the recording (seen in Figs. 1a, b, 2b and Figs S1c, f).

**Simultaneous behavior tracking and calcium imaging**. We used a customized optical imaging system to track a single worm and simultaneously image its intestinal and neuronal cells[37]. Our closed-loop setup consisted of a high-speed CMOS camera for locomotion and behavior tracking and a two-camera neural-circuit imaging subsystem. Using a fast-tracking algorithm, the behavior tracking system drove a two-axis piezo stage to realize worm-motion compensation, enabling rapid and stable cellular-resolution neural-circuit imaging.

The transgenic lines kyIs772 and kyIs777 in wild-type and various genetic background (Table 1) were used for imaging. Single L3-L4 stage animals were transferred to customized NGM plates seeded with a thin layer of OP50 lawn. GCaMP and mCherry signals in freely-moving animals were recorded with a 20x/0.75 objective, and two sCMOS cameras (Hamamatsu, ORCA-Flash4.0 V3) at a frame rate of 24 Hz. For kyIs772 where only GCaMP was imaged (Fig. 5 and S4), the frame rate was 8 Hz.

**Data analysis**

*Electrophysiology*. Statistics, curve fitting, and graphing were conducted using OriginPro 2020b (OriginLab). Action potential spikes for averaging were manually selected for the largest peak amplitude for each preparation. Measurements of resting membrane potential, threshold, peak amplitude, AHP, and spike width were made by hand with the measure function in Fitmaster (Heka). Electrophysiological recording traces shown in all figures were re-sampled at 2.5 or 1 kHz for reducing file size. Boltzmann function for IV curve fitting was $y = \frac{A_1 - A_2}{1 + e^{(x - x_0)/x_1}} + A_2$.

**Behavior and calcium imaging**. Analysis of behavior and calcium imaging data was performed using custom MATLAB scripts. To perform worm shape analysis, the centerline of each worm was first extracted frame-by-frame using a centerline localization algorithm, and then equally divided into 49 segments[37]. The position of one of the VD/DD motor neurons was used as a landmark to divide the worm

centerline into anterior and posterior halves. Then the length of each half was calculated separately to distinguish pBoc and aBoc. For intestinal imaging, the normal to the centerline at each point partitioned the worm intestine into equally-spaced segments. The intestinal GCaMP or mCherry fluorescence intensity from anterior to posterior was defined by the average fluorescence intensity within each segment. For neuronal imaging, the GCaMP or mCherry fluorescence intensities of AVL and DVB soma were defined as the average fluorescence intensity of a region of interest (ROI) in the soma region. The calcium activity (R) was calculated as the ratio of GCaMP to mCherry fluorescence intensities and were normalized by its minimum ($R_0$) as $(R - R_0)/R_0$. The timing of neuronal firing was defined as the time when its calcium dynamic begins to rise from the baseline. The timing of expulsion was defined as the start of an obvious enteric muscle contraction that can be observed by the naked eye.

**Statistics and reproducibility**. No statistical method was used to predetermine sample size. The experimental animals were randomly selected from a group of animals maintained in the same culture condition. The investigators were not blinded to allocation during experiments and outcome assessment. For electrophysiology recording, only patch clamps with a seal resistance above 1 GΩ and uncompensated series resistance below 100 MΩ were accepted for further statistical analysis. For calcium imaging, the cycles in which the worm was always fully inside

## Table 1 List of worm strains.

| Stain ID | Genotype |
|---|---|
| EG1285 | lin-15B(n765) X; oxIs12 [Punc-47::GFP::lin-15(+)] X |
| CX17773 | exp-2(sa26ad1426) V; lin-15B(n765) X; oxIs12 X |
| CX17868 | +/eT1 III; exp-2(sa26)/eT1[let-(n886)] V; lin-15B(n765) X; oxIs12 X |
| CX17774 | shl-1(ok1168) IV; lin-15B(n765) X; oxIs12 X |
| CX17889 | shl-1(ok1168) IV; exp-2(sa26ad1426) V; lin-15B(n765) X; oxIs12 X |
| CX17776 | unc-2(e55) X; lin-15B(n765) X; oxIs12 X |
| CX18359 | unc-2(lj1) X; lin-15B(n765) X; oxIs12 X |
| CX18364 | kyEx6593 [Punc-47::synthetic intron::pBluescript SK(-)::unc-2 cDNA with synthetic intron::N-GFP::unc-54 UTR complete @ 100 ng/ul + Punc-122 (coelomocytes)::dsRed @ 10 ng/ul]; unc-2(lj1) X; lin-15B(n765) X; oxIs12 X |
| CX17777 | egl-19(n582) IV; lin-15B(n765) X; oxIs12 X |
| CX17848 | oxIs608 [Punc-47::mCherry] V; cca-1(ad1650) X |
| CX17849 | egl-19(n582) IV; oxIs608 V; cca-1(ad1650) X |
| CX17890 | nca-2(gk5) III; nca-1(gk9) IV; lin-15B(n765) X; oxIs12 X |
| CX17891 | unc-2(zf35) X; lin-15B(n765) X; oxIs12 X |
| CX17902 | egl-19(n582) IV; unc-2(e55) X; lin-15B(n765) X; oxIs12 X |
| CX18004 | kyIs772 [Punc-47mini::GCaMP6f @ 50 ng/ul + Pinx-16::GCaMP6f @ 5 ng/ul + Pceh-63::GCaMP6f @ 25 ng/ul] |
| CX18153 | kyIs777 [Punc-47::GCaMP6f::SL2::mCherry::let858 3′UTR @ 30 ng/ul + Punc-47mini::GCaMP6f::SL2::mCherry::unc-54 3′ UTR @ 50 ng/ul + Pinx-16::GCaMP6f @ 5 ng/ul] II |
| CX18176 | kyIs777 II; inx-1(tm3524) X |
| CX18360 | kyEx6589 [Punc-47::inx-1a cDNA @ 25 ng/ul + Punc-122 (coelomocytes)::dsRed @ 10 ng/ul]; kyIs777 II; inx-1(tm3524) X |
| CX18316 | kyIs777 II; exp-2(sa26ad1426) V |
| CX18362 | KyEx6591 [Punc-47::exp-2 cDNA::SL2-mCherry::let858 3′UTR @ 89 ng/ul + Punc-122 (coelomocytes)::dsRed @ 10 ng/ul]; exp-2(sa26ad1426) V; lin-15B(n765) X; oxIs12 X |

## Table 2 Plasmid list.

| Plasmid ID | Gene name |
|---|---|
| pQL1 | Punc-47mini(196 bp)::GCaMP6f::unc-54 3′UTR |
| pQL2 | Pinx-16::GCaMP6f::unc-54 3′UTR |
| pQL3 | Pceh-63::GCaMP6f::unc-54 3′UTR |
| pQL6 | Punc-47::GCaMP6f::SL2-mCherry::let858 3′UTR |
| pQL7 | Punc-47mini::GCaMP6f::SL2::mCherry::unc-54 3′UTR |
| pQL11 | Punc-47::exp-2 cDNA::SL2-mCherry::let858 3′UTR |
| pQL13 | Punc-47::synthetic intron:: pBluescript SK(-):: unc-2 cDNA with synthetic intron::N-GFP::unc-54 UTR |
| pHW175 | Punc-47::inx-1a cDNA |

the field of view and never in a coiled shape were accepted for further statistical analysis.

**Reporting summary**. Further information on research design is available in the Nature Research Reporting Summary linked to this article.

## Data availability

All electrophysiology and imaging data presented in this study have been deposited in the Mendeley Data under CC BY 4.0 license (https://doi.org/10.17632/33mnszcbh8.1). Source data are provided with this paper.

## Code availability

Customized codes used in this study has been deposited in GitHub.
AVL-modeling code: https://github.com/taolab302/AVL-Modeling
Imaging analysis code: https://github.com/taolab302/DMP-Imaging-Processing

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

## Acknowledgements

The completion of this work could not have been possible without the encouragement and advisement of Cori Bargmann. We thank Cori for the discussion throughout this project and her comments on the manuscript. We thank the *Caenorhabditis* Genetics Center (CGC), Ian Hope, and Erik Jorgensen for strains; Derek Sieburth for strains, plasmids, and comments on the manuscript; Margaret Ebert for assistance on transgenics; Friederike Buck and Philip Kidd for comments on the manuscript. This work was supported by the Natural Science Foundation of China through grants 32020103007 (J.J., Y.S., R.Z., and L.T.), 21927813 (J.J., Y.S., and L.T.), and 31771147 (H.L. and L.T.), NSF CRCNS grant (2113120) and the Kavli NSI Pilot Grant to Q.L. This work was also supported by the laboratory of Cori Bargmann at the Rockefeller University and the Chan Zuckerberg Initiative.

## Author contributions

Q.L. conceived the project and designed the experiments. L.T. oversaw the live-imaging experiments and modeling. Q.L. carried out electrophysiological experiments and

constructed the imaging strains used in this work. J.J., Y.S., H.L., and Q.L. carried out live-imaging experiments. J.J. analyzed the behavior tracking and imaging data. Y.S. and R.Z. developed the biophysical models. Q.L. wrote the paper with input from all authors.

## Competing interests

The authors declare no competing interests.
