## [Peer review file · Nature Communications]

REVIEWER COMMENTS

Reviewer #1 (Remarks to the Author):

Jiang et al investigate the electrophysiological properties of two enteric neurons AVL and DVB and reveal a number of illuminating features about how their unique electrophysical properties may contribute to the *C. elegans* defecation circuit. Specifically they show that AVL and DVB exhibit all-or-none activity, described as “spikes” or “action potentials.” They use mutants and modeling to identify the specific channels and genes that generate these spikes, coordinate activity and regulate timing.

The quality of the work is high. The manuscript is clearly the result of very challenging work. The manuscript is presented logically and is well written. The work will be of interest to the *C. elegans* community and also to the community of systems neuroscientists. Below I highlight a number of areas that could use clarification, but none call into doubt the core claims of the manuscript nor do they detract from my overall enthusiasm for the work.

- (Line 368) Authors claim that AVL’s “averaged AHP” lasted for over 40 sec, matching the duration of the intercycle, but Fig 6H only shows two examples and no statistics or is reported (at least none that I could find). Please provide evidence and summary statistics to back up this claim, or temper it. As this claim has relevance for interpreting AVL’s role in regulating defecation timing.

- (Line 333) Authors claim that DVB spikes were “not detected before AVL spikes (Fig 6G),” and that the average is 215 ms but the scale on Figure 6G is inadequate to resolve the dots all scrunched together, or even for a reader to confirm that the dots are indeed all positive. Please present a clearer visualization.

- In the variable duration current clamp experiments in Fig 2A, the duration of current injection is always longer than the action potential. Is there a reason that shorter current injections are not explored? Could those experiments be informative?

- (Line 210) Figures 1AB and presumably also 2B (although its unclear) show that a negative current pulse is delivered for approx. 0.5 s prior to the main current clamp experiment. Please clarify the role of this. Is the cell’s response during this pulse relevant or informative?

- The authors show a particularly strong result with the double mutant *egl-19* and *unc-2*. If the authors have them, it would be nice to see behavioral results from this mutant. Do they still have successful expulsion events? (Note this is not strictly required for publication, but if the authors have this information they should report it.)

- In some instances, the mutant analysis presented here leaves open the possibility that an effect that is presumed to be cell-autonomous could in principle be a network effect. For example, it would seem conceivable that the negative spike in AVL could be, for example, from a reciprocal inhibitory connection. Can this easily be ruled out? Please comment on this possibility. For example, a cell-specific genetic rescue of *exp-2* using the drivers in this paper would be one way to address this issue, but it may not be necessary or practical.

Superficial:

- Figure 6 legend and label fonts are too small are completely illegible when reading a paper copy.

- Figure 3C is missing an axis label. I assume this is voltage, even though the vertical axis in nearby panels A & B refers to current.

Reviewer #2 (Remarks to the Author):

The authors build up on their previous discovery of action potentials in the AWA sensory neurons and show the presence of action potentials in AVL and DVB motor neurons. Their data indicates that AVL fires unusual compound action potentials. The data also shows that AVL functions through UNC-2 and EXP-2 ion channels. The data from this paper is especially interesting as AVL records a negative spike that appears to be dependent on the inward rectifier potassium channel, EXP-2. Further, AVL propagates the action potential all along its axon from head to tail to activate DVB through an innexin and this circuit is required for egg expulsion. This manuscript is very well written and enhances our knowledge about the firing patterns and molecular mechanisms of AVL and DVB function. This will be of great interest to researchers studying aspects of *C. elegans* neuroscience as well as researchers

interested in understanding the underpinning of voltage-gated calcium channels and inward rectifier potassium channels in action potentials.

The data shown largely supports the claims of the authors and the methods used by the authors appear to be well thought out and detailed. I have concerns listed below that the authors could address. These are largely control experiments that may strengthen their conclusions.

Abstract:

The second last sentence needs rewriting as it is unclear.

Introduction and discussion:

The introduction and discussion are well written. The introduction does an excellent job of laying out the thinking and hypothesis driving this paper. My only suggestion would be to cite the Mellem, JE et al 2008, Nature Neuroscience paper as part of the introduction.

Results:

1. The authors should add all of their unpublished control data indicated in the manuscript. At the least they should add the data on recording from RIS and DVC upon current injection. These are very important control experiments that indicate the specificity of action potentials in AVL and DVB.
2. The statistical tests used in some figures are not clear, can the authors make sure that the legends carry the statistical tests used.
3. Although the authors have nicely shown that UNC-2 is required for the action potentials through AVL, rescue experiments may enhance their current data.
4. The authors show through calcium imaging that the action potential from AVL propagates along the AVL axon to DVB to allow for synchronous firing of AVL and DVB. Have the authors tried blocking the action potential movement through the axon through mutants. As an example, loss of kinesin heavy chain has been previously shown to impair action potentials in Drosophila neurons (Gho et al., 1992; Science).
5. The data showing that long AHP is present in exp-2 mutants would be a nice control that can be added to figure 6.

Reviewer #3 (Remarks to the Author):

This manuscript identifies novel all or nothing calcium-mediated action potentials in two enteric GABAergic neurons in *C. elegans* hermaphrodites, using technically difficult techniques and validation with calcium recording. The AVL neuron displays compound action potentials with a depolarizing calcium spike and a hyperpolarizing potassium spike. Using single loss and single gain of function mutants in *unc-2* the authors demonstrate a role for the *unc-2* gene in mediating the depolarization, while a similar analysis using *exp-2* mutants suggest a role in mediating the negative spike. The authors perform analysis of AVL and DVB neuron activity in behaving animals together with calcium imaging, which demonstrates synchronization of the two neurons. This synchronization is shown to be required for expulsion/defecation behavior and relies on the *inx-2* gap junction gene. This work is novel and exciting in that it identifies action potentials in two additional neurons in *C. elegans*, while the presence of action potentials in *C. elegans* has been debated for many years. Furthermore these are observed in GABAergic motor neurons (interneurons), extending beyond action potentials previously reported in a sensory neuron (AWA). The sophisticated behavioral work demonstrates the importance of the action potentials in these neurons to behavior. The work included in the manuscript is impressive and the overall impact of the work to the *C. elegans* field and beyond is important and novel. However, there are a number of claims that are overstated given the data shown, and some claims that could be more strongly supported by additional experiments, which would improve the manuscript. The modeling experiments are compelling and interesting, although this is beyond the expertise of this referee to comment on the rigor of this work.

Major comments

1. While there are clear roles for *unc-2* and *exp-2* in the action potentials of AVL, the claims of the importance of *unc-2* seem overstated (a “primary role” (line 184)). Use of the term “complete action potential” (lines 176-177) is further misleading as *unc-2* (lf) mutants appear to show AVL depolarization, even if often lacking negative spike. While there is certainly a reduction in the depolarization, it seems likely that other genes are must also be important (ie. *egl-21*). Inclusion of expression data or mention of whether *unc-2* or *exp-2* genes are expressed in AVL, or relevant tissues/neurons upstream or downstream of AVL would be informative for readers.
2. The manuscript would be strengthened by including recordings of DVB in *unc-2* and/or *egl-19* mutants for comparison with the AVL characterization.
3. Figure 5A,B panels are hard to interpret as the images are very overexposed/saturated (and would benefit from inclusion of a scale bar). Although the authors claim that DVC is “irrelevant to this study”(Figure 5 legend), it would strengthen the results to include the unpublished recording data of DVC (cited in line 125) and even analysis of the calcium reporter in DVC as a negative control for the DVB calcium imaging in Figure 5 – this would also demonstrate accurate identification of the two close neurons.
4. The methods section makes it unclear if the calcium imaging and behavior experiments were done in adults or in L3-L4 worms – which is an important point for readers to understand. “571 All animals used

in this study were young adult hermaphrodites (~3-days old)", "642 (Table 1) were used for imaging. L3-L4 stage animals were transferred to customized NGM". This should be clarified in the results and methods sections.

5. While the *inx-1* results in Figure 6 are very compelling, they would be strengthened by confirming localization of the *inx-1* gene in AVL/DVB using transgenic rescue as done in the cited Choi et al paper. Although Choi et al localized the *inx-1* gene to AVL/DVB for behavioral analysis, this study should confirm this localization and requirement of *inx-1* for spike entrainment between the neurons as well.

6. Since the *exp-2(gf)* mutants have narrower spikes, it would be interesting and perhaps further strengthen the behavioral analysis to look at whether timing of defecation cycle is altered in *exp-2(gf)* heterozygous mutants as may be predicted by spike width correlation observed in *exp-2(lf)*. Line 383: "The averaged delay between the two consecutive expulsions in *exp-2(lf)* was 430 ± 155 ms (n=6), which corresponds well with the averaged spike width of AVL action potential (394.2 ± 40.0 ms, n =17) (Fig 7E). This result suggests that negative spikes in AVL may prevent excessive expulsions by 386 terminating calcium signals in AVL and/or DVB in a timely manner"

Minor comments:

1. Parallel structure of the text and order of Figure 1 would improve clarity — *dvb* is explained first and then *avl* in the text but displayed in the opposite order in the figure

2. Figure 2G should have indications of non-significance in the first and third graph to increase ease of reader interpretation

3. Figures would be improved with consistent color coding – red and blue based on neuron to match the traces– ex. figure 6 D & E

4. Comments on other potential genes that may be contributing to AVL/DVB synchrony other than *inx-1* would be helpful since *inx-1* does not cause complete loss. Line 339: "43% of cycles in *inx-1(lf)* compared to 100% in WT"

5. These two statements appear contradictory, Line 342: "DVB firing spikes alone without AVL firing was never observed in either WT (n=17 cycles) or *inx-1* (n=14 cycles)" and Line 373: "In the *inx-1* mutant, elimination of this inhibitory entrainment force by the removal of the gap junction could allow DVB to fire at its own pace"

Reviewer #1

Jiang et al investigate the electrophysiological properties of two enteric neurons AVL and DVB and reveal a number of illuminating features about how their unique electrophysical properties may contribute to the *C. elegans* defecation circuit. Specifically they show that AVL and DVB exhibit all-or-none activity, described as “spikes” or “action potentials.” They use mutants and modeling to identify the specific channels and genes that generate these spikes, coordinate activity and regulate timing.

The quality of the work is high. The manuscript is clearly the result of very challenging work. The manuscript is presented logically and is well written. The work will be of interest to the *C. elegans* community and also to the community of systems neuroscientists. Below I highlight a number of areas that could use clarification, but none call into doubt the core claims of the manuscript nor do they detract from my overall enthusiasm for the work.

- (Line 368) Authors claim that AVL’s “averaged AHP” lasted for over 40 sec, matching the duration of the intercycle, but Fig 6H only shows two examples and no statistics or is reported.

Response: The panels on the right in Fig 6H (now as 6G) are actually averaged traces from multiple recordings ($n=7$, the gray shade is SEM). We have made the figures larger and the legends clearer.

- (Line 333) Authors claim that DVB spikes were “not detected before AVL spikes (Fig 6G),” and that the average is 215 ms but the scale on Figure 6G is inadequate to resolve the dots all crunched together, or even for a reader to confirm that the dots are indeed all positive. Please present a clearer visualization.

Response: We changed the scale and data presentation on Fig 6G (now as Fig 7D) for better visualization.

- In the variable duration current clamp experiments in Fig 2A, the duration of current injection is always longer than the action potential. Is there a reason that shorter current injections are not explored? Could those experiments be informative?

Response: Shorter current injections were explored. An example trace was added to Fig S1C. Short current injections resulted in truncated positive calcium spike due to early onset of negative spike, consistent with our biophysical model shown in Fig 3C.

- (Line 210) Figures 1AB and presumably also 2B (although its unclear) show that a negative current pulse is delivered for approx. 0.5 s prior to the main current clamp experiment. Please clarify the role of this. Is the cell’s response during this pulse relevant or informative?

Response: The negative “prepulse” is part of the protocol to measure the input resistance before each step to make sure that the seal of patch remains unchanged during the stimulation. Unfortunately, the time when the prepulse was applied has not been consistent in different sets of experiments throughout the study. That’s why some of traces show it some not (when the

prepulse was applied before time 0). We added this explanation in the Methods section (page 20).

- The authors show a particularly strong result with the double mutant *egl-19* and *unc-2*. If the authors have them, it would be nice to see behavioral results from this mutant. Do they still have successful expulsion events? (Note this is not strictly required for publication, but if the authors have this information they should report it.)

Response: Although the reviewer stated that this is not strictly required for publication, we still performed said experiments to strengthen our model. In *egl-19;unc-2* double mutant, the detectable expulsion events and anterior body contractions (aBoc) were essentially abolished while posterior body contractions (pBoc) were intact. This observation is consistent with our electrophysiological results and the literature (pBoc is resulted from direct activation of body-wall muscles by proton released from the gut). A representative example trace is shown in Fig S1H.

- In some instances, the mutant analysis presented here leaves open the possibility that an effect that is presumed to be cell-autonomous could in principle be a network effect. For example, it would seem conceivable that the negative spike in AVL could be, for example, from a reciprocal inhibitory connection. Can this easily be ruled out? Please comment on this possibility. For example, a cell-specific genetic rescue of *exp-2* using the drivers in this paper would be one way to address this issue, but it may not be necessary or practical.

Response: It is theoretically possible that the negative spike in AVL could result from a reciprocal inhibitory connection in an EXP-2 dependent fashion. We tested cell-autonomous effects as the reviewer suggested by injecting *Punc-47* promoter driven *exp-2*cDNA in *exp-2(lf)* mutant to rescue *exp-2* in AVL, DVB and other GABA neurons. We found that the negative spike can be partially restored in *exp-2* rescued animals identified by positive co-injection markers (n=5). All 5 rescuing traces were shown in Fig S2C. Interestingly, the amplitudes of both calcium spikes and negative potassium spikes in the *exp-2* rescued animals were reduced, which may result from increased leaking currents due to EXP-2 overexpression and/or mis-localization of EXP-2 in AVL. We further simulated the effect of increased leaking current in our biophysical model and were able to reproduce comparable reduction of spike amplitudes in both directions (Fig S2D). These results suggest that the ratio of UNC-2/EXP-2 and/or the subcellular localization of EXP-2 relative to UNC-2 is important for the coupling of positive Ca²⁺ spike and negative K⁺ spike. Although we still cannot completely rule out the possibility that a *Punc-47+* neuron forming reciprocal inhibitory connection with AVL may theoretically be responsible, such hypothetical connection cannot be from DVB because no EXP-2 currents were detected in DVB (as shown in Fig S2B) for generating negative spikes, the fact that EXP-2-mediated repolarization-activated K⁺ currents were recorded in AVL and our mathematical single-cell AVL model could simulate precisely the shape and time course of the negative spike strongly suggest a cell-autonomous effect. Above results and discussion were added to the revised manuscript.

- Figure 6 legend and label fonts are too small are completely illegible when reading a paper copy.

- Figure 3C is missing an axis label. I assume this is voltage, even though the vertical axis in nearby panels A & B refers to current.

Response: We have made changes accordingly.

Reviewer #2

The authors build up on their previous discovery of action potentials in the AWA sensory neurons and show the presence of action potentials in AVL and DVB motor neurons. Their data indicates that AVL fires unusual compound action potentials. The data also shows that AVL functions through UNC-2 and EXP-2 ion channels. The data from this paper is especially interesting as AVL records a negative spike that appears to be dependent on the inward rectifier potassium channel, EXP-2. Further, AVL propagates the action potential all along its axon from head to tail to activate DVB through an innexin and this circuit is required for egg expulsion. This manuscript is very well written and enhances our knowledge about the firing patterns and molecular mechanisms of AVL and DVB function. This will be of great interest to researchers studying aspects of *C. elegans* neuroscience as well as researchers interested in understanding the underpinning of voltage-gated calcium channels and inward rectifier potassium channels in action potentials. The data shown largely supports the claims of the authors and the methods used by the authors appear to be well thought out and detailed. I have concerns listed below that the authors could address. These are largely control experiments that may strengthen their conclusions.

Abstract:

The second last sentence needs rewriting as it is unclear.

Response: We changed the sentence to: "These results suggest that three biophysical properties of two electrically coupled enteric motoneurons, including synchronous firing of calcium action potentials in AVL and DVB, time-locked hyperpolarizing potassium spike in AVL, and long-lasting afterhyperpolarization in AVL, are important for the rhythmic expulsion behavior in *C. elegans*."

Introduction and discussion:

The introduction and discussion are well written. The introduction does an excellent job of laying out the thinking and hypothesis driving this paper. My only suggestion would be to cite the Mellem, JE et al 2008, Nature Neuroscience paper as part of the introduction.

Response: Said citation was added in page 3.

Results:

1. The authors should add all of their unpublished control data indicated in the manuscript. At the least they should add the data on recording from RIS and DVC upon current injection. These are very important control experiments that indicate the specificity of action potentials in AVL and DVB.

Response: Examples of RIS and DVC current-clamp recording traces have been added in Fig S1A and S1B.

2. The statistical tests used in some figures are not clear, can the authors make sure that the legends carry the statistical tests used.

Response: We have double checked all figure legends and made sure statistical tests are clearly labeled.

3. Although the authors have nicely shown that UNC-2 is required for the action potentials through AVL, rescue experiments may enhance their current data.

Response: To address the reviewer's concern, we first tested a second *unc-2* loss-of-function allele *lj1* and got consistent results (Fig S1D). Secondly, we rescued *unc-2(lj1)* in AVL by injecting UNC-2cDNA under *Punc-47* promoter and the compromised action potentials were fully rescued (Fig S1E).

4. The authors show through calcium imaging that the action potential from AVL propagates along the AVL axon to DVB to allow for synchronous firing of AVL and DVB. Have the authors tried blocking the action potential movement through the axon through mutants. As an example, loss of kinesin heavy chain has been previously shown to impair action potentials in *Drosophila* neurons (Gho et al., 1992; Science).

Response: The experiment that the reviewer suggested would be good additional evidence of action potential propagation, we however did not perform this experiment due to time and reagent constraints.

5. The data showing that long AHP is present in *exp-2* mutants would be a nice control that can be added to figure 6.

Response: We added the result that the reviewer suggested (Fig 6G, right). It now nicely shows that the fast negative spike and prolonged AHP are genetically separable.

Reviewer #3 (Remarks to the Author):

This manuscript identifies novel all or nothing calcium-mediated action potentials in two enteric GABAergic neurons in *C. elegans* hermaphrodites, using technically difficult techniques and validation with calcium recording. The AVL neuron displays compound action potentials with a depolarizing calcium spike and a hyperpolarizing potassium spike. Using single loss and single gain of function mutants in *unc-2* the authors demonstrate a role for the *unc-2* gene in mediating the depolarization, while a similar analysis using *exp-2* mutants suggest a role in mediating the negative spike. The authors perform analysis of AVL and DVB neuron activity in behaving animals together with calcium imaging, which demonstrates synchronization of the two neurons. This synchronization is shown to be required for expulsion/defecation behavior and relies on the *inx-2* gap junction gene. This work is novel and exciting in that it identifies action potentials in two additional neurons in *C. elegans*, while the presence of action potentials in *C. elegans* has been debated for many years. Furthermore these are observed in GABAergic motor neurons (interneurons), extending beyond action potentials previously reported in a sensory neuron

(AWA). The sophisticated behavioral work demonstrates the importance of the action potentials in these neurons to behavior. The work included in the manuscript is impressive and the overall impact of the work to the *C. elegans* field and beyond is important and novel. However, there are a number of claims that are overstated given the data shown, and some claims that could be more strongly supported by additional experiments, which would improve the manuscript. The modeling experiments are compelling and interesting, although this is beyond the expertise of this referee to comment on the rigor of this work.

Major comments

1. While there are clear roles for *unc-2* and *exp-2* in the action potentials of AVL, the claims of the importance of *unc-2* seem overstated (a “primary role” (line 184)). Use of the term “complete action potential” (lines 176-177) is further misleading as *unc-2* (lf) mutants appear to show AVL depolarization, even if often lacking negative spike. While there is certainly a reduction in the depolarization, it seems likely that other genes are must also be important (ie. *egl-21*). Inclusion of expression data or mention of whether *unc-2* or *exp-2* genes are expressed in AVL, or relevant tissues/neurons upstream or downstream of AVL would be informative for readers.

Response: Based on recent system-wide RNAseq data base (CeNGEN), both *unc-2* and *exp-2* genes are expressed in AVL (Citation of CeNGEN are added). As we responded to the other reviewers: we performed cell-specific rescuing experiments using Punc-47 promotor driven UNC-2cDNA or EXP-2cDNA and found that the defective depolarization for either positive spike or negative spike can be rescued by expressing UNC-2 or EXP-2 in GABA neurons (Fig S1D and S2C). These new results suggest that the functions of UNC-2 and EXP-2 are necessary in generating spikes in a cell-autonomous fashion. Other genes that function together with UNC-2 and EXP-2 may also be required.

2. The manuscript would be strengthened by including recordings of DVB in *unc-2* and/or *egl-19* mutants for comparison with the AVL characterization.

Response: We agree with the reviewer that examining various calcium channel mutants in DVB would be informative. However, dissecting DVB in the tail (especially in calcium channel mutant with smaller body size) is much more challenging than dissecting AVL. Therefore we did not perform the suggested experiments due to time and technical constrains and instead refer to DVB calcium imaging data in calcium channel mutants in the literature.

3. Figure 5A,B panels are hard to interpret as the images are very overexposed/saturated (and would benefit from inclusion of a scale bar). Although the authors claim that DVC is “irrelevant to this study”(Figure 5 legend), it would strengthen the results to include the unpublished recording data of DVC (cited in line 125) and even analysis of the calcium reporter in DVC as a negative control for the DVB calcium imaging in Figure 5 – this would also demonstrate accurate identification of the two close neurons.

Response: We replaced Fig 5A with an images of better quality and added a scale bar. We also reduced the exposure of Fig 5B while keeping the thin axon visible. Fluorescence in DVC is now analyzed as a negative control shown in Fig 5C. Example of DVC current-clamp recording traces is added to Fig S1A.

4. The methods section makes it unclear if the calcium imaging and behavior experiments were

done in adults or in L3-L4 worms – which is an important point for readers to understand. “571 All animals used in this study were young adult hermaphrodites (~3-days old)”, “642 (Table 1) were used for imaging. L3-L4 stage animals were transferred to customized NGM“. This should be clarified in the results and methods sections.

Response: Young adult worms were only used in electrophysiological experiments. L3-L4 stage animals were used for imaging and behavioral tracking. This oversight has been corrected.

5. While the *inx-1* results in Figure 6 are very compelling, they would be strengthened by confirming localization of the *inx-1* gene in AVL/DVB using transgenic rescue as done in the cited Choi et al paper. Although Choi et al localized the *inx-1* gene to AVL/DVB for behavioral analysis, this study should confirm this localization and requirement of *inx-1* for spike entrainment between the neurons as well.

Response: We agree with the reviewer that transgenic rescue of *inx-1* gene in AVL/DVB for spike entrainment between the neurons would strengthen our model. Therefore, we reached out to Derek Sieburth and obtained the *Punc-47* promoter driven INX-1cDNA that was used in the Choi et al study. We injected the rescuing constructs into our-dual color imaging strain and selected rescued animals based on co-injection marker for imaging. The result was quite satisfying - expressing INX-1 in AVL and DVB largely rescued most of the synchronized firing and expulsion defects in *inx-1(lf)*. These new results were shown in Fig 6D,E, Fig 7B-D, and Fig S5D.

6. Since the *exp-2(gf)* mutants have narrower spikes, it would be interesting and perhaps further strengthen the behavioral analysis to look at whether timing of defecation cycle is altered in *exp-2(gf)* heterozygous mutants as may be predicted by spike width correlation observed in *exp-2(lf)*. Line 383: “The averaged delay between the two consecutive expulsions in *exp-2(lf)* was 430 ± 155 ms (n=6), which corresponds well with the averaged spike width of AVL action potential (394.2 ± 40.0 ms, n =17) (Fig 7E). This result suggests that negative spikes in AVL may prevent excessive expulsions by 386 terminating calcium signals in AVL and/or DVB in a timely manner”

Response: We attempted the experiment suggested by the reviewer but without success. The main obstacle was that the *exp-2(gf)* allele we were using has severe behavioral and morphological defects and crossing it with the dual-color imaging strain made it even worse. The few viable putative heterozygous cross progenies we managed to obtain all had very weak neural GCaMP signals to be examined reliably. We and our collaborators are in the process of making point-mutations in EXP-2 to make new viable *exp-2(gf)* alleles in order to study EXP-2 channel in details, which we feel is beyond the scope of the current paper and warrants a follow-up study.

Minor comments:

1. Parallel structure of the text and order of Figure 1 would improve clarity — *dvb* is explained first and then *avl* in the text but displayed in the opposite order in the figure

2. Figure 2G should have indications of non-significance in the first and third graph to increase ease of reader interpretation

3. Figures would be improved with consistent color coding – red and blue based on neuron to match the traces– ex. figure 6 D & E

Response: We made all the changes suggested by the reviewer.

4. Comments on other potential genes that may be contributing to AVL/DVB synchrony other than *inx-1* would be helpful since *inx-1* does not cause complete loss. Line 339: “43% of cycles in *inx-1(lf)* compared to 100% in WT”

Response: We are not sure whether other innexins also contribute to AVL/DVB synchrony or not based this observation. There is little doubt that AVL can induce synchronized firing in DVB through gap junctions, but without gap junctions, DVB may still fire out-of-phase APs on its own, possibly through slow-diffusing NLP-40 peptides (discussed in page 16). As can be seen from AVL-DVB spike delay in Fig 7D, all remaining DVB firing are out of phase in *inx-1(lf)* which constitutes the 43% cycle with DVB firing alone mentioned here.

5. These two statements appear contradictory, Line 342: “DVB firing spikes alone without AVL firing was never observed in either WT (n=17 cycles) or *inx-1* (n=14 cycles)” and Line 373: “In the *inx-1* mutant, elimination of this inhibitory entrainment force by the removal of the gap junction could allow DVB to fire at its own pace”

Response: Similar to the point above, by “DVB firing at its own pace” we did mean DVB firing alone without AVL firing, but asynchronous DVB and AVL firing with variable delays shown in Fig 7D. We realized that this point was not clearly conveyed in the original manuscript and have made further clarification in the revision.

REVIEWERS' COMMENTS

Reviewer #1 (Remarks to the Author):

The revisions have addressed my concerns.

Reviewer #2 (Remarks to the Author):

The authors have responded satisfactorily to all my concerns. I support the publication of this manuscript in Nature Communications.

Reviewer #3 (Remarks to the Author):

The authors have addressed my concerns, and done a good job responding to comments from myself and the other reviewers. Clarification and additional experiments have made the manuscript stronger, and have increased my enthusiasm for this important and well done work.